# Faster Non-asymptotic Convergence for Double Q-learning

**Lin Zhao**
National University of Singapore
elezhli@nus.edu.sg

**Huaqing Xiong**
The Ohio State University
xiong.309@osu.edu

**Yingbin Liang**
The Ohio State University
liang.889@osu.edu

## Abstract

Double Q-learning (Hasselt, 2010) has gained significant success in practice due to its effectiveness in overcoming the overestimation issue of Q-learning. However, the theoretical understanding of double Q-learning is rather limited. The only existing finite-time analysis was recently established in (Xiong et al., 2020), where the polynomial learning rate adopted in the analysis typically yields a slower convergence rate. This paper tackles the more challenging case of a constant learning rate, and develops new analytical tools that improve the existing convergence rate by orders of magnitude. Specifically, we show that synchronous double Q-learning attains an $\epsilon$-accurate global optimum with a time complexity of $\tilde{\Omega}\left(\frac{\ln D}{(1-\gamma)^7\epsilon^2}\right)$, and the asynchronous algorithm achieves a time complexity of $\tilde{\Omega}\left(\frac{L}{(1-\gamma)^7\epsilon^2}\right)$, where $D$ is the cardinality of the state-action space, $\gamma$ is the discount factor, and $L$ is a parameter related to the sampling strategy for asynchronous double Q-learning. These results improve the existing convergence rate by the order of magnitude in terms of its dependence on all major parameters $(\epsilon, 1-\gamma, D, L)$. This paper presents a substantial step toward the full understanding of the fast convergence of double-Q learning.

## 1 Introduction

Double Q-learning, proposed in Hasselt (2010), is a widely used model-free reinforcement learning (RL) algorithm in practice for learning an optimal policy (Zhang et al., 2018a,b; Hessel et al., 2018). Compared to the vanilla Q-learning proposed in Watkins and Dayan (1992), double Q-learning uses two Q-estimators with their roles randomly selected at each iteration, respectively for estimating the maximal Q-function value and updating the Q-function. In this way, the overestimation of the Q-function in the vanilla Q-learning can be effectively mitigated, especially when the reward is random or prone to errors (Hasselt, 2010; Hasselt et al., 2016; Xiong et al., 2020). Moreover, double Q-learning has been shown to have the desired performance in both finite state-action space setting (Hasselt, 2010) and infinite setting (Hasselt et al., 2016) where it successfully improved the performance of deep Q-network (DQN), and thus inspired more variants (Zhang et al., 2017; Abed-alguni and Ottom, 2018) subsequently.

In parallel to its empirical success in practice, the theoretical convergence properties of double Q-learning has also been explored. Its asymptotic convergence was first established in Hasselt (2010). The asymptotic mean-square error for double Q-learning was studied in Weng et al. (2020b) under the assumption that the algorithm converges to a unique optimal policy. More recently, in Xiong et al. (2020), the finite-time convergence rate has been established for double Q-learning with a polynomial learning rate $\alpha = 1/t^\omega, \omega \in (0,1)$. Under such a choice of the learning rate, Xiong et al. (2020) showed that double Q-learning attains an $\epsilon$-accurate optimal Q-function at a time complexity approaching to but never reaching $\Omega(\frac{1}{\epsilon^2})$ at the cost of an asymptotically large exponent of $\frac{1}{1-\gamma}$.

Table 1: Comparison of time complexity for synchronous and asynchronous double Q-learning.

| SyncDQ | Stepsize | Time complexity† | |
|---|---|---|---|
| Xiong et al. (2020) | $\frac{1}{t^\omega}, \omega \in (\frac{1}{3}, 1)$ | $\omega = 1 - \eta \to 1$ | $\omega = 6/7$ |
| | | $\tilde{\Omega}\left(\frac{1}{\epsilon^{2+\eta}} \vee \left(\frac{1}{1-\gamma}\right)^{\frac{1}{\eta}}\right)$ | $\tilde{\Omega}\left(\frac{1}{(1-\gamma)^7}\left(\frac{1}{\epsilon^{3.5}} \vee \left(\ln\frac{1}{1-\gamma}\right)^7\right)\right)$ |
| **This work** | $\epsilon^2(1-\gamma)^6$ | $\tilde{\Omega}\left(\frac{1}{\epsilon^2}\right)$ | $\tilde{\Omega}\left(\frac{1}{(1-\gamma)^7\epsilon^2}\right)$ |

| AsyncDQ | Stepsize | Time complexity † | | |
|---|---|---|---|---|
| Xiong et al. (2020) | $\frac{1}{t^\omega}, \omega \in (\frac{1}{3}, 1)$ | $\omega = 1 - \eta \to 1$ | $\omega = 6/7$ | $\omega = 2/3$ |
| | | $\tilde{\Omega}\left(\frac{1}{\epsilon^{2+\eta}} \vee \left(\frac{1}{1-\gamma}\right)^{\frac{1}{\eta}}\right)$ | $\tilde{\Omega}\left(\frac{1}{(1-\gamma)^7}\left(\frac{1}{\epsilon^{3.5}} \vee \left(\ln\frac{1}{1-\gamma}\right)^7\right)\right)$ | $\tilde{\Omega}\left(\frac{L^6(\ln L)^{1.5}}{(1-\gamma)^9\epsilon^3}\right)$ |
| **This work** | $\epsilon^2(1-\gamma)^6$ | $\tilde{\Omega}\left(\frac{1}{\epsilon^2}\right)$ | $\tilde{\Omega}\left(\frac{1}{(1-\gamma)^7\epsilon^2}\right)$ | $\tilde{\Omega}\left(\frac{L}{(1-\gamma)^7\epsilon^2}\right)$ |

† The choices $\omega \to 1, \omega = \frac{6}{7}$, and $\omega = \frac{2}{3}$ optimize the dependence of time complexity on $\epsilon, 1 - \gamma$, and $L$ in Xiong et al. (2020), respectively. In addition, we denote $a \vee b = \max\{a, b\}$.

However, a polynomial learning rate typically does not offer the best possible convergence rate, as having been shown for RL algorithms that a *constant learning rate* achieves better convergence bounds (Beck and Srikant, 2012; Bhandari et al., 2018; Li et al., 2020). Therefore, a natural question arises as follows:

*Can a constant learning rate improve the convergence rate of double Q-learning by orders of magnitude? If yes, does it also improve the dependence of the convergence rate on other important parameters of the Markov decision process (MDP) such as the discount factor and the cardinality of the state-action space?*

In this paper, we develop novel analysis techniques and provide affirmative answers to the above questions.

### 1.1 Our contributions

This paper establishes sharper finite-time bounds for double Q-learning with constant learning rates, which improve the existing bounds by orders of magnitude. Our result hence encourages to apply the constant stepsize in practice to attain better convergence performance.

For synchronous double Q-learning, where all state-action pairs are visited at each iteration, we show that with constant learning rates $\alpha_t \equiv \alpha \in (0, 1)$, the algorithm converges to an $\epsilon$-accurate global optimum with a time complexity of $\tilde{\Omega}\left(\frac{\ln D}{(1-\gamma)^7\epsilon^2}\right)$, where $\gamma$ is the discount factor and $D = |\mathcal{S}||\mathcal{A}|$ is the cardinality of the finite state-action space. As a comparison, for the $\epsilon$-dominated regime (with relatively small $\gamma$), we show that double Q-learning attains an $\epsilon$-accurate optimal Q-function with a time complexity of $\Omega(\frac{1}{\epsilon^2})$, whereas the result in Xiong et al. (2020) (see Table 1) does not exactly reach $\Omega(\frac{1}{\epsilon^2})$ and its approaching to such an order ($\eta := 1 - \omega \to 0$) is at an additional cost of an asymptotically large exponent on $\frac{1}{1-\gamma}$. For the $(1 - \gamma)$-dominated regime, our result improves on that in Xiong et al. (2020) (which has been optimized in the dependence on $1 - \gamma$ in Table 1 by $\mathcal{O}\left(\left(\ln\frac{1}{1-\gamma}\right)^7\right)$.

For asynchronous double Q-learning, where only one state-action pair is visited at each iteration via a single sample trajectory, we show that the algorithm attains an $\epsilon$-accurate global optimum with a time complexity of $\tilde{\Omega}\left(\frac{L}{(1-\gamma)^7\epsilon^2}\right)$, where $L$ denotes the covering time (see (15)), and depends on the sampling strategy. As illustrated in Table 1, our result improves upon that in Xiong et al. (2020) order-wisely in terms of its dependence on $\epsilon$ and $1 - \gamma$ as well as on $L$ by at least $\mathcal{O}\left(L^5\right)$.

Technically, our finite-time analysis approach here is very different yet more direct than the techniques in Xiong et al. (2020). Our goal is still to bound the convergence error via a pair of nested stochastic approximation (SA) recursions, where the outer SA captures the *learning error dynamics* between

one Q-estimator and the global optimum and the inner SA captures the error propagation between the two Q-estimators. Rather than constructing two block-wisely decreasing bounds for the nested SAs as in Xiong et al. (2020), which appears challenging if not infeasible under the constant learning rate, we devise new analysis techniques to directly bound both the inner and outer error dynamics either per iteration (for synchronous sampling) or per frame of constant iterations (for asynchronous sampling). We then treat the output of the inner SA as an noise term in the outer SA, and combine the two bounds to establish the finite-time error bound of the learning error.

## 1.2 Related work

Due to large number of studies in Q-learning, we only review the most relevant ones on the double Q-learning and the vanilla Q-learning under tabular/function approximation settings.

**Theory on double Q-learning:** Double Q-learning was proposed and proved to converge asymptotically in Hasselt (2010). In Weng et al. (2020b), the authors explored the properties of mean-square errors for double Q-learning both in the tabular case and with linear function approximation, under the assumption that a unique optimal policy exists and the algorithm can converge. The most relevant work to this paper is Xiong et al. (2020), which established the first finite-time convergence rate for tabular double Q-learning with a polynomial learning rate. This paper provides sharper finite-time convergence bounds for double Q-learning under a constant learning rate, which requires a different analysis approach.

**Tabular Q-learning and convergence under various learning rates:** Proposed in Watkins and Dayan (1992) under finite state-action space, Q-learning has aroused great interest in its theoretical study. Its asymptotic convergence has been established in Tsitsiklis (1994); Jaakkola et al. (1994); Borkar and Meyn (2000); Melo (2001); Lee and He (2020) by requiring the learning rates to satisfy $\sum_{t=0}^{\infty} \alpha_t = \infty$ and $\sum_{t=0}^{\infty} \alpha_t^2 < \infty$. Another line of research focuses on the finite-time analysis of Q-learning under different choices of the learning rates. Szepesvári (1998) captured the first convergence rate of Q-learning using a linear learning rate (i.e., $\alpha_t = \frac{1}{t}$). Under similar learning rates, Even-Dar and Mansour (2003) provided finite-time results for both synchronous and asynchronous Q-learning with a convergence rate being exponentially slow as a function of $\frac{1}{1-\gamma}$. Another popular choice is the polynomial learning rate which has been studied for synchronous Q-learning in Wainwright (2019b) and for both synchronous/asynchronous Q-learning in Even-Dar and Mansour (2003). With this learning rate, however, the convergence rate still has a gap with the lower bound (Azar et al., 2013). To handle this, a more sophisticated rescaled linear learning rate was introduced for synchronous Q-learning (Wainwright, 2019b,c; Chen et al., 2020) and asynchronous Q-learning (Qu and Wierman, 2020), and thus yields a better convergence rate. The finite-time bounds for Q-learning were also given for constant learning rate (Beck and Srikant, 2012; Chen et al., 2020; Li et al., 2020). In this paper, we focus on the constant learning rate and obtain sharper finite-time bounds for double Q-learning.

**Q-learning with function approximation:** When the state-action space is considerably large or even infinite, the Q-function is usually approximated by a class of parameterized functions. In such a case, Q-learning has been shown not to converge in general (Baird, 1995). Strong assumptions are typically needed to establish the convergence of Q-learning with linear function approximation (Bertsekas and Tsitsiklis, 1996; Melo et al., 2008; Zou et al., 2019; Chen et al., 2021b; Du et al., 2019; Yang and Wang, 2019; Jia et al., 2019; Weng et al., 2020a, 2021) or neural network approximation (Cai et al., 2019; Xu and Gu, 2020). A two time-scale variation of Q-learning with linear function approximation was proposed in (Carvalho et al., 2020) which provably converges with probability 1. The convergence analysis of double Q-learning with function approximation raises new technical challenges and is an interesting topic for our future study.

## 2 Preliminaries on Double Q-learning

We consider a Markov decision process (MDP) over a finite state space $\mathcal{S}$ and a finite action space $\mathcal{A}$ with the total cardinality given by $D := |\mathcal{S}||\mathcal{A}|$. The transition kernel of the MDP is given by $\mathbb{P} : \mathcal{S} \times \mathcal{A} \times \mathcal{S} \to [0,1]$ denoted as $\mathbb{P}(\cdot|s,a)$. We assume a *random* reward function, denoted by $R_t : \mathcal{S} \times \mathcal{A} \times \mathcal{S} \mapsto [0,1]$, and its expectation is denoted by $\mathbb{E}[R_t(s,a,s')] = R_{sa}^{s'}$. Note that the analysis of this paper easily applies to reward functions of different ranges by scaling. A policy $\pi := \pi(\cdot|s)$ captures the conditional probability distribution over the action space given state $s \in \mathcal{S}$.

For a policy $\pi$, we define the Q-function (i.e., the state action value function) $Q^\pi \in \mathbb{R}^{|\mathcal{S}| \times |\mathcal{A}|}$ as

$$Q^\pi(s,a) = \mathop{\mathbb{E}}_{\substack{a_t \sim \pi(\cdot|s_t) \\ s'_t \sim \mathbb{P}(\cdot|s_t,a_t)}} \sum_{t=1}^\infty \gamma^t R_{s_t,a_t}^{s'_t},$$

with $(s_1, a_1) = (s, a)$, and $\gamma \in (0, 1)$ is the discount factor.

Both the vanilla Q-learning (Watkins and Dayan, 1992) and double Q-learning (Hasselt, 2010) aim to find the optimal Q-function $Q^*$ which is the unique fixed point of the Bellman operator $\mathcal{T}$ (Bertsekas and Tsitsiklis, 1996) given by

$$\mathcal{T}Q(s,a) = \mathbb{E}_{s' \sim \mathbb{P}(\cdot|s,a)} \left[ R_{sa}^{s'} + \gamma \max_{a' \in \mathcal{A}} Q(s', a') \right]. \tag{1}$$

Note that the Bellman operator $\mathcal{T}$ is $\gamma$-contractive which satisfies $\|\mathcal{T}Q - \mathcal{T}Q'\| \le \gamma \|Q - Q'\|$ under the supremum norm $\|Q\| := \max_{s,a} |Q(s,a)|$.

**Double Q-learning:** The idea of double Q-learning is to keep two Q-tables (i.e., Q-function estimators) $Q^A$ and $Q^B$, and randomly choose one Q-table to update at each iteration based on the Bellman operator computed from the other Q-table. Let $\{\beta_t\}_{t \ge 1}$ be a sequence of i.i.d. Bernoulli random variables satisfying $\mathbb{P}(\beta_t = 0) = \mathbb{P}(\beta_t = 1) = 0.5$. At each time $t$, $\beta_t = 0$ indicates that $Q^B$ is updated, and otherwise $Q^A$ is updated. The update at time $t \ge 1$ can be written as,

$$\begin{cases} Q_{t+1}^A(s,a) = (1 - \hat{\alpha}_t(s,a)\beta_t)Q_t^A(s,a) + \hat{\alpha}_t(s,a)\beta_t \left( R_t + \gamma Q_t^B(s', a^*) \right), \\ Q_{t+1}^B(s,a) = (1 - \hat{\alpha}_t(s,a)(1-\beta_t)) Q_t^B(s,a) + \hat{\alpha}_t(s,a)(1 - \beta_t) \left( R_t + \gamma Q_t^A(s', b^*) \right), \end{cases} \tag{2}$$

for all $(s,a) \in \mathcal{S} \times \mathcal{A}$, $a^* = \arg\max_{a \in \mathcal{A}} Q^A(s', a)$, and $b^* = \arg\max_{a \in \mathcal{A}} Q^B(s', a)$. The difference between the (a)synchronous sampling is captured by the learning rate

$$\hat{\alpha}_t(s,a) := \begin{cases} \alpha, & \text{for synchronous version} \\ \alpha \tau_t(s,a), & \text{for asynchronous version} \end{cases} \tag{3}$$

where $\tau_t(s,a) = \mathbb{1}_{\{(s_t,a_t)=(s,a)\}}$ is an indicator function. It can be seen that different from synchronous double Q-learning which updates all $(s,a)$-pairs at each iteration, the asynchronous algorithm samples only one state-action pair to update a chosen Q-estimator. Particularly, in the synchronous version, $s'$ is sampled independently between iterations following $s' \sim \mathbb{P}(\cdot|s,a)$ for all $(s,a)$, whereas in the asynchronous version, $s'$ are consecutive sample transitions from one observed trajectory (i.e., Markovian sampling). Finally, we note that in (2) the rewards $R_t$ for both updates of $Q_{t+1}^A$ and $Q_{t+1}^B$ are the same copy of $R_t(s,a,s')$.

As indicated by the update rules (2), at each iteration only one of the two Q-tables is randomly chosen to be updated. This chosen Q-table generates a greedy optimal action, and the other Q-table is used for estimating the corresponding Bellman operator (or evaluating the greedy action) to update the chosen table. Specifically, if $Q^A$ is chosen to be updated, we use $Q^A$ to obtain the optimal action $a^*$ and then estimate the corresponding Bellman operator using $Q^B$ to update $Q^A$. As shown in Hasselt (2010), $\mathbb{E}[Q^B(s', a^*)]$ is likely smaller than $\mathbb{E} \max_a [Q^A(s', a)]$, where the expectation is taken over the randomness of the reward for the same $(s, a, s')$ tuple. Such a two-estimator framework adopted by double Q-learning can effectively reduce the overestimation.

Without loss of generality, we assume that $Q^A$ and $Q^B$ are initialized with the same value (usually both are all-zero tables in practice). For (a)synchronous double Q-learning, it can be easily shown that either Q-estimator is uniformly bounded by $V_{\max} := \frac{1}{1-\gamma}$ throughout the learning process (see, for example Xiong et al. (2020)). Specifically, for either $i \in \{A, B\}$, we have $\|Q_t^i\| \le V_{\max}$ and $\|Q_t^i - Q^*\| \le V_{\max}$ for all $t \ge 1$, which will be useful in the finite-time analysis in the sequel.

## 3 Synchronous Double Q-learning

In this section, we start with modeling the error dynamics to be nested SAs, which will be useful to understand the nature of double Q-learning algorithms. Then we provide the finite-time results for synchronous double Q-learning, followed by a proof sketch to understand analysis insights.

## 3.1 Characterization of the Error Dynamics

In the following, we represent the (a)synchronous double Q-learning algorithms as a pair of nested SA recursions, where the outer SA recursion captures the error dynamics between the Q-estimator and the global optimum $Q^*$, and the inner SA captures the error propagation between the two Q-estimators which enters into the outer SA as a noise term. Such a characterization enjoys various useful properties and prompts a structured way of finite-time analysis for both sampling cases.

**Outer SA:** Denote the iteration error by $r_t(s,a) = Q_t^A(s,a) - Q^*(s,a)$ and define the empirical Bellman operator $\widehat{\mathcal{T}}_t Q(s,a) := R_t(s,a,s') + \gamma \max_{a' \in \mathcal{A}} Q(s',a')$. Then we can have for all $t \geq 1$ (see Appendix A),

$$r_{t+1}(s,a) = (1 - \tilde{\alpha}_t(s,a))r_t(s,a) + \tilde{\alpha}_t(s,a)\big(\mathcal{G}_t(r_t)(s,a) + \varepsilon_t(s,a) + \gamma\nu_t(s',a^*)\big), \quad (4)$$

where $\varepsilon_t := \widehat{\mathcal{T}}_t Q^* - Q^*$, $\nu_t := Q_t^B - Q_t^A$, $\mathcal{G}_t(r_t) := \widehat{\mathcal{T}}_t Q_t^A - \widehat{\mathcal{T}}_t Q^* = \widehat{\mathcal{T}}_t(r_t + Q^*) - \widehat{\mathcal{T}}_t Q^*$, and the equivalent learning rate $\tilde{\alpha}_t(s,a) := \begin{cases} \alpha\beta_t, & \text{for synchronous version} \\ \alpha\beta_t\tau_t(s,a), & \text{for asynchronous version} \end{cases}$. Note that it is by design that we use the same sampled reward $R_t$ in both $\widehat{\mathcal{T}}_t Q^*$ and $\widehat{\mathcal{T}}_t Q_t^A$ in the definition of $\mathcal{G}_t(r_t)$.

These newly introduced variables have several important properties. First of all, the noise term $\{\varepsilon_t\}_t$ is a sequence of i.i.d. random variables satisfying $\mathbb{E}\varepsilon_t = \mathbb{E}[\widehat{\mathcal{T}}_t Q^*] - Q^* = \mathcal{T}Q^* - Q^* = \mathbf{0} \in \mathbb{R}^D$. In particular, for the asynchronous case, we have $\tau_t^T \varepsilon_t = \varepsilon_t(s_t, a_t)$ which is a Markovian noise (process) for $t \geq 1$, where $\tau_t, \varepsilon_t \in \mathbb{R}^D$. Furthermore, it can be shown that (see Appendix A)

$$\|\varepsilon_t\| \leq V_{\max} = \frac{1}{1-\gamma}. \quad (5)$$

Moreover, it is easy to show that $\|\mathcal{G}_t(r_t)\| \leq \gamma \|r_t\|$, which follows from the contractive property of the empirical Bellman operator given the same next state. We shall say that $\mathcal{G}_t$ is *quasi-contractive* in the sense that the $\gamma$-contraction inequality holds only with respect to the origin $r_t = \mathbf{0}$.

**Inner SA:** We further characterize the dynamics of $\nu_t = Q_t^B - Q_t^A$ as an SA recursion (see Appendix A):

$$\nu_{t+1}(s,a) = (1 - \hat{\alpha}_t(s,a))\nu_t(s,a) + \hat{\alpha}_t(s,a)\left(\mathcal{H}_t(\nu_t)(s,a) + \mu_t(s,a)\right), \quad (6)$$

where $\hat{\alpha}_t(s,a)$ is defined in (2), $\mathcal{H}_t := \mathbb{E}\left(H_t | \mathcal{F}_t\right)$, $\mu_t := H_t - \mathcal{H}_t$, and $H_t$ is defined in (18) in Appendix A. It can be shown that $\mathcal{H}_t$ is quasi-contractive satisfying $\|\mathcal{H}_t(\nu_t)\| \leq \frac{1+\gamma}{2}\|\nu_t\|$, and $\{\mu_t\}_{t \geq 1}$ is a martingale difference sequence with respect to the filtration $\mathcal{F}_t$ defined by $\mathcal{F}_1 = \{\emptyset, \Omega\}$ where $\Omega$ denotes the underlying probability space and for $t \geq 2$,

$$\mathcal{F}_t = \begin{cases} \sigma\left(\{s_k\}, \{R_{k-1}\}, \beta_{k-1}, 2 \leq k \leq t\right), & \text{for synchronous version,} \\ \sigma\left(s_k, a_k, R_{k-1}, \beta_{k-1}, 2 \leq k \leq t\right), & \text{for asynchronous version.} \end{cases} \quad (7)$$

We note that for synchronous sampling, $\{s_k\}$ and $\{R_{k-1}\}$ are respectively the collections of sampled next states and the sampled rewards for each $(s,a)$ pair; and for asynchronous sampling, $\{(s_k, a_k, s_{k+1})\}_{k \geq 2}$ are consecutive sample transitions from one observed trajectory.

Now our goal is to provide the finite-time bound for the nested SA recursions (4) and (6). Note that $\beta_t$ (present in the outer SA) captures the random switching behavior of double Q-learning, and plays an important role in connecting the error dynamics of inner and outer SAs. The error dynamics of the inner SA, which has a unique contractive operator, affects the overall convergence via its coupling to the error dynamics of the outer SA.

In the sequel, we present our main result of the finite-time error bounds and consequently the time complexity of synchronous double Q-learning. We also make a thorough comparison with existing results to highlight our significant improvement.

## 3.2 Finite-time Analysis

For the synchronous double Q-learning, at each iteration, one of the two Q-tables is chosen and all its state-action pairs are sampled and updated. In Theorem 1, we present a finite-time error bound on the learning error $\|r_t\| = \|Q_t^A - Q^*\|$, based on which we further derive the time complexity in Corollary 1. We then provide a proof sketch of Theorem 1 in Section 3.4. Its complete proof is deferred to Appendix B.

**Theorem 1.** *Consider the synchronous double Q-learning in (2) with a constant learning rate $\alpha_t \equiv \alpha \in (0,1)$ and $\gamma \in (0,1)$. Then there exist some universal constant $c > 0$, such that with probability at least $1 - \delta$, the learning error $r_t = Q_t^A - Q^*$ satisfies*

$$\|r_{t+1}\| \leq h^t \|r_1\| + \frac{c}{(1-\gamma)^3} \sqrt{\alpha \ln \frac{2D}{\delta}}, \tag{8}$$

*for all $t \geq 1$, where $h := 1 - \frac{1-\gamma}{2}\alpha$.*

The following insights of the convergence of double-Q learning can be drawn from (8). The first term on the RHS of (8) shows that the initial error decays linearly with respect to the number of iterations (in logarithmic scale). The second term is a constant error which can be controlled by the size of the learning rate. By decreasing the learning rate $\alpha$, this error vanishes at an order of $\mathcal{O}(\sqrt{\alpha})$. This bound can be used to guide the choice of stepsizes according to the desired learning accuracy levels. Note that there is trade-off between the initial error decay rate and the constant learning error: a smaller stepsize $\alpha$ yields a smaller learning error, but leads to a larger $h$ that is closer to $1$ and thus slower decaying of the initial error.

With the above result, we are ready to derive the following time-complexity result, which characterizes the number of iterations required to achieve a statistically meaningful finite-time error bound using synchronous double-Q learning. Recall that with the reward $R_t \in [0,1]$, the Q-function falls within the range $[0, \frac{1}{1-\gamma}]$. Therefore, it is reasonable to consider $\epsilon \in (0, \frac{1}{1-\gamma}]$.

**Corollary 1.** *For any $\epsilon \in (0, \frac{1}{1-\gamma}]$, the time complexity for synchronous double Q-learning to achieve an $\epsilon$-accurate optimal Q-function (i.e., $\|r_T\| \leq \epsilon$) with probability at least $1 - \delta$ is given by*

$$T(\epsilon, \gamma, \delta, D) = \tilde{\Omega}\left(\frac{\ln \frac{D}{\delta}}{(1-\gamma)^7 \epsilon^2}\right).$$

*Proof.* We first show that if $\alpha \leq \frac{\epsilon^2(1-\gamma)^6}{4c^2 \ln \frac{2D}{\delta}}$ and $T \geq \frac{2}{(1-\gamma)\alpha} \ln \frac{2}{\epsilon(1-\gamma)}$, then $\|r_{T+1}\| \leq \epsilon$. To proceed, using Theorem 1, the first term on the right hand side (RHS) of (8) can be bounded as

$$h^T \|r_1\| \overset{(i)}{\leq} \frac{\epsilon(1-\gamma)}{2} \|r_1\| \overset{(ii)}{\leq} \frac{\epsilon}{2}, \tag{9}$$

where (i) follows from the inequality $1 + x \leq \exp(x)$ and (ii) follows from the bound $\|r_1\| \leq \frac{1}{1-\gamma}$.

Applying the upper bound of $\alpha$ to the second term on the RHS of (8), we obtain $\frac{c}{(1-\gamma)^3} \sqrt{\alpha \ln \frac{2D}{\delta}} \leq \frac{\epsilon}{2}$, which combining with (9) proves the initial claim.

Then, further applying the bound on $\alpha$ in the lower bound of $T$, we obtain the desired time complexity bound $T \geq \frac{8c^2 \ln \frac{2D}{\delta}}{\epsilon^2(1-\gamma)^7} \ln \frac{2}{\epsilon(1-\gamma)}$, and $\tilde{\Omega}$ denotes the omission of logarithmic factors in the corollary. $\square$

**Comparison with existing results**: We compare our result with the time complexity of synchronous double Q-learning provided in Xiong et al. (2020), which is given by

$$T = \Omega\left(\left(\frac{1}{(1-\gamma)^6 \epsilon^2} \ln \frac{D}{(1-\gamma)^7 \epsilon^2}\right)^{\frac{1}{\omega}} + \left(\frac{1}{1-\gamma} \ln \frac{1}{(1-\gamma)^2 \epsilon}\right)^{\frac{1}{1-\omega}}\right), \tag{10}$$

where $\omega \in (\frac{1}{3}, 1)$. We consider the following two major regimes. (a) For the $\epsilon$-**dominated regime** (with relatively small $\gamma$), clearly the result in (10) does not achieve the order of $\Omega\left(\ln D/\epsilon^2\right)$ whereas our result does. Further, its approaching to such an order ($\omega \to 1$ in Table 1) is also at an additional cost of an asymptotically large exponent on $\frac{1}{1-\gamma}$ (see the last term of (10)). (b) For the $(1-\gamma)$-**dominated regime**, the dependence on $1 - \gamma$ can be optimized by taking $\omega = \frac{6}{7}$ in (10), compared to which our result achieves an improvement by a factor of $\mathcal{O}\left(\left(\ln \frac{1}{1-\gamma}\right)^7\right)$ (see Table 1).

Comparing to the lower bound obtained in Azar et al. (2013), which is $T = \Omega(\frac{1}{(1-\gamma^3)\epsilon^2} \log \frac{D}{\delta})$ after translating to finite-time complexity under synchronous sampling, we see that the orders of dependence on $D$ and $\delta$ are tight. The order of $\varepsilon$ is tight up to a logarithm factor. The only major gap appears in the dependence on $1 - \gamma$.

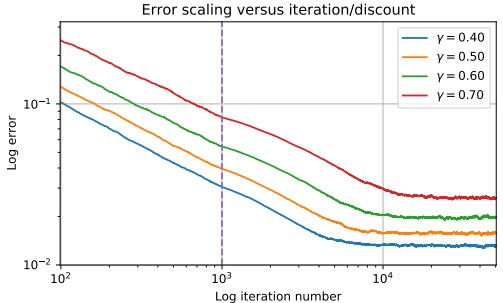
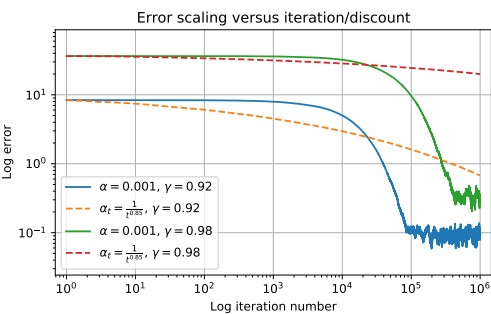

Figure 1: Convergence rate under constant step-size (after $10^3$ iterations)

Figure 2: Convergence comparison using different stepsizes

## 3.3 Numerical Experiment

To illustrate our theoretical results, we apply the synchronous double Q-learning to an MDP adapted from (Wainwright, 2019b) by modifying the deterministic rewards there to be uniformly distributed over $\{R_{sa} - 0.5, R_{sa} + 0.5\}$, where $R_{sa} \in \{0, 1\}$ is the expected reward at $(s, a)$. We set the initial conditions as $Q^A = Q^B = \mathbf{1.0}$ with appropriate dimensions. Figure 1 illustrates the convergence rate of synchronous double Q-learning under constant stepsize with different discount values. Since the decay of the initial error usually dominates the learning process, it is not easy to observe the slower convergence rate $T = \Omega(\frac{1}{\epsilon^2})$ when only using constant stepsize. To better illustrate the latter, we first apply the rescaled linear stepsizes ($\alpha_t = \frac{2}{2+(1-\gamma)t}$) at the beginning to reduce the initial errors. Then from $t = 10^3$, we switch to a constant stepsize of $\alpha = 0.001$. The plotted curves are averaged over 1000 independent runs. It can be observed that the ratio in the middle region under constant stepsize is about $-0.5$, which is consistent with our analysis of $T = \Omega(\frac{1}{\epsilon^2} \log \frac{1}{\epsilon}) = \tilde{\Omega}(\frac{1}{\epsilon^2})$ (see Corollary 1). Besides, the non-diminishing learning errors are also captured by our Theorem 1.

We also compare the convergence against a polynomial stepsize ($\alpha_t = \frac{1}{t^{0.85}}$) using the same MDP as in the last example with the same initialization of the Q functions. The results are shown in Fig. 2 which are averaged over 10 independent runs. Under the same discount factor, a constant stepsize generally yields faster convergence than a polynomial stepsize due to the linear decay rate (see Theorem 1). Convergence becomes slower when increasing the discount factor under both stepsizes, but polynomial stepsize is more susceptible to the change of discount factor than constant stepsize.

## 3.4 Proof Sketch of Theorem 1

We first give a high-level idea about the difference between our approach and that in Xiong et al. (2020). Our central goal here is to bound the convergence error via a pair of nested SA recursions, where the outer SA captures the learning error dynamics between one Q-estimator and the global optimum, and the inner SA captures the error propagation between the two Q-estimators. Xiong et al. (2020) constructed two block-wisely decreasing bounds for the nested SAs and characterized only a block-wise convergence. Such an approach is rather complicated, and does not appear to extend easily to the constant learning rate case. In contrast, we take a very different yet more direct approach here. We devise new analysis techniques to directly bound both the inner and outer error dynamics per iteration. We then treat the output of the inner SA as a noise term in the outer SA, and combine the two bounds to establish the finite-time error bound of the learning error.

Specifically, our proof of Theorem 1 consists of five steps.

**Step I: Deriving a template finite-time bound applicable to both SAs.** We focus on a general SA with a quasi-contractive operator, which captures the interconnection structure between the outer and inner SAs of double Q-learning via a few noise and error terms. We provide a finite-time bound for such an SA by unfolding the SA recursively and carefully combining those noise terms into several smaller recursions. Such a bound involves the coupling of the outer and inner SA error terms, and is critical for further deriving a sharper finite-time bound for the synchronous Double-Q learning. We present this bound in Proposition 1 in Appendix C.

**Step II: Bounding outer SA dynamics $\mathbb{E}\|r_t\|$ by inner SA dynamics $\mathbb{E}\|\nu_t\|$.** We apply Proposition 1 to the error dynamics of $r_t$ in (4), and take the expectation to obtain

$$\mathbb{E}\|r_{t+1}\| \leq h^t \|r_1\| + \frac{\gamma}{2}\alpha \sum_{k=1}^{t} h^{t-k} (\mathbb{E}\|W_k\| + \mathbb{E}\|\nu_k\|) + \mathbb{E}\|W_t\|, \tag{11}$$

where $h = 1 - \frac{1-\gamma}{2}\alpha$ and $W_{t+1} = (1 - \tilde{\alpha}_t)W_t + \tilde{\alpha}_t\varepsilon_t$, with initialization $W_1 = \mathbf{0}$.

The bound in (11) captures the coupling between the nested SAs of double Q-learning, where $\nu_t$ of the inner SA enters as a tracking error term into the bound on $r_t$ of the outer SA. To further bound the outer SA error, we need to handle both $\|\nu_t\|$ and $\|W_t\|$.

**Step III: Bounding $\mathbb{E}\|W_t\|$.** To bound $\mathbb{E}\|W_t\|$, we first construct an $\mathcal{F}_t$-martingale sequence

$$\tilde{W}_i := \left(1 - \frac{\alpha}{2}\right)^{t-i} W_i, \quad \text{for } 1 \leq i \leq t$$

where it can be shown that $\tilde{W}_t = W_t$ and $\tilde{W}_1 = \mathbf{0}$. Following this way, for each $t > 1$, we construct $\tilde{W}_t$ as the martingale surrogate of $W_t$. Equipped with such a property, and further applying the Azuma-Hoeffding inequality (see Lemma 3) and a Gaussian function integration technique (see Lemma 4), we show that

$$\mathbb{E}\|W_{t+1}\| \leq 2\tilde{D}V_{\max}\sqrt{\alpha},$$

where $\tilde{D} = 2\sqrt{\ln 2D} + \sqrt{\pi}$ and $V_{\max}$ is defined in (5). The complete results are presented in Proposition 2.

**Step IV: Bounding inner SA dynamics $\mathbb{E}\|\nu_t\|$.** The inner SA captures the difference between two Q-estimators, and is an error existing specifically in double Q-learning. Bounding $\mathbb{E}\|\nu_t\|$ is one of the key steps to handle the coupling between two nested SAs.

Note that the $\nu_t$-recursion (6) also satisfies the SA condition in Proposition 1. Then we have

$$\mathbb{E}\|\nu_t\| \leq \tilde{\gamma}\alpha \sum_{k=1}^{t-1} h^{t-k-1}\mathbb{E}\|M_k\| + \mathbb{E}\|M_t\|, \tag{12}$$

where $\tilde{\gamma} := \frac{1+\gamma}{2}$ and $M_{t+1} = (1-\alpha_t)M_t + \alpha_t\mu_t$, with initialization $M_1 = \mathbf{0}$. To bound $\mathbb{E}\|M_t\|$, we construct the martingale surrogate of $M_t$, for each $t \geq 1$. Specifically, we construct the $\mathcal{F}_t$-martingale sequence $\{\tilde{M}_i\}_{1\leq i\leq t+1}$ as $\tilde{M}_i := (1-\alpha)^{t-i+1}M_i$ with $\tilde{M}_{t+1} = M_{t+1}$ and $\mathbb{E}\left(\tilde{M}_1\right) = \mathbf{0}$. Then by further characterizing the boundedness of the martingale, applying the Azuma-Hoeffding inequality, and the Gaussian function integration technique in Lemma 4, we obtain in Proposition 3 that

$$\mathbb{E}\|M_{t+1}\| \leq 2\tilde{D}V_{\max}\sqrt{2\alpha}.$$

Finally, we substitute the bound of $\mathbb{E}\|M_t\|$ into (12) to have

$$\mathbb{E}\|\nu_t\| \leq \frac{6\tilde{D}V_{\max}\sqrt{2\alpha}}{1-\gamma}. \tag{13}$$

The above bound indicates that the inner SA error is uniformly bounded by a constant which can be controlled by the learning rate. Thus, the learning rate will play an important role to guarantee the overall convergence by ensuring such an error to asymptotically vanish when entering into the outer SA error as in (11).

**Step V: Deriving overall finite-time complexity.** The last step is to "instantiate" the bounds of $\mathbb{E}\|W_t\|$ and $\mathbb{E}\|\nu_t\|$ in (11), from which we have for any $t \geq 1$,

$$\mathbb{E}\|r_{t+1}\| \leq h^t \|r_1\| + \frac{6\tilde{D}\sqrt{2\alpha}}{(1-\gamma)^3}. \tag{14}$$

The above expectation bound is equivalent to the high probability bound on $\|r_t\|$ by replacing $\ln 2D$ with $c\ln\frac{2D}{\delta}$ for some universal constant $c > 0$ (see for example Wainwright (2019b)). We then readily have that (8) holds with probability at least $1 - \delta$.

# 4 Asynchronous Double Q-learning

In this section, we first develop the finite-time analysis for asynchronous double Q-learning, and then provide a proof sketch for the main theorem.

## 4.1 Finite-time Analysis

Differently from the synchronous algorithm, at each iteration asynchronous double Q-learning updates only one state-action pair for a randomly chosen Q-estimator. The sampled state-action pairs come from a single trajectory of the underlying MDP. Thus the statistical properties of the MDP plays an important role in the finite-time analysis.

We next make the following standard assumption, which is widely used in the studies of asynchronous Q-learning (Paulin, 2015; Qu and Wierman, 2020; Li et al., 2020).

**Assumption 1.** *The Markov chain induced by the stationary behavior policy $\pi$ is uniformly ergodic.*

We further introduce the following notations that will be used in the analysis. We denote by $\mu_\pi$ the stationary distribution of the behavior policy over the state-action space $\mathcal{S} \times \mathcal{A}$ and denote $\mu_{\min} := \min_{(s,a) \in \mathcal{S} \times \mathcal{A}} \mu_\pi(s, a)$. Intuitively, it suggests that the smaller $\mu_{\min}$ is, the more iterations it takes to visit all state-action pairs in a single trajectory. Alternatively, we define the so-called *covering number* in the following, which is the number of iterations required to cover all state-action pairs at least once with the probability of at least one-half:

$$L = \min \left\{ t : \min_{(s_1, a_1) \in \mathcal{S} \times \mathcal{A}} \mathbb{P}(\mathcal{B}_t | (s_1, a_1)) \geq \frac{1}{2} \right\}, \tag{15}$$

where $\mathcal{B}_t$ denotes the event that all state-action pairs have been visited at least once in $t$ iterations.

In addition, the ergodicity assumption indicates that the distribution of samples will approach to the stationary distribution $\mu_\pi$ at the so-called mixing rate, defined as

$$t_{\mathrm{mix}} = \min \left\{ t : \max_{(s_1, a_1) \in \mathcal{S} \times \mathcal{A}} d_{\mathrm{TV}} \left( P^t(\cdot | (s_1, a_1)), \mu_\pi \right) \leq \frac{1}{4} \right\}, \tag{16}$$

where $P^t(\cdot | (s_1, a_1))$ is the distribution of $(s_t, a_t)$ given the initial pair $(s_1, a_1)$, and $d_{\mathrm{TV}}(\mu, \nu)$ is the variation distance between two distributions $\mu, \nu$.

Next, we provide the finite-time convergence result for asynchronous double Q-learning in the following theorem. We provide a proof sketch in Section 4.2 and give the full proof in Appendix F.

**Theorem 2.** *Let $\gamma \in (0, 1), \delta \in (0, \frac{1}{7}), \epsilon \in (0, \frac{1}{1-\gamma}]$ and suppose Assumption 1 holds. Consider asynchronous double Q-learning with a constant learning rate $\alpha_t \equiv \alpha = \frac{c_1}{\ln \frac{DT}{\delta}} \min \left\{ (1 - \gamma)^6 \epsilon^2, \frac{1}{t_{\mathrm{mix}}} \right\}$ with some universal constant $c_1 > 0$. Then asynchronous double Q-learning learns an $\epsilon$-accurate optimum, i.e., $\left\| Q_t^A - Q^* \right\| \leq \epsilon$, with probability at least $1 - 7\delta$ given the number $T$ of iterations satisfies*

$$T = \tilde{\Omega} \left( \left( \frac{1}{\mu_{\min} \epsilon^2 (1-\gamma)^7} + \frac{t_{\mathrm{mix}}}{\mu_{\min}(1-\gamma)} \right) \ln \frac{1}{\epsilon(1-\gamma)^2} \right),$$

*where $t_{\mathrm{mix}}$ is defined in (16).*

The above time complexity result is given in terms of the mixing time. To facilitate the comparison with the existing results, we further characterize it in terms of the covering number, which can be obtained from standard arguments for Q-learning (see (Li et al., 2020)).

**Corollary 2.** *Under the same conditions of Theorem 2, consider a constant learning rate $\alpha_t \equiv \alpha = \frac{c_2}{\ln \frac{DT}{\delta}} \min \left\{ (1 - \gamma)^6 \epsilon^2, 1 \right\}$ with some constant $c_2 > 0$. Then asynchronous double Q-learning can learn an $\epsilon$-accurate optimum, i.e., $\left\| Q_T^A - Q^* \right\| \leq \epsilon$, with probability at least $1 - 7\delta$ if the number $T$ of iterations satisfies*

$$T = \tilde{\Omega} \left( \frac{L}{\epsilon^2 (1-\gamma)^7} \ln \frac{1}{\epsilon(1-\gamma)^2} \right),$$

*where $L$ is defined in (15).*

**Comparison with existing results**: We compare our result with the time complexity of asynchronous double Q-learning in Xiong et al. (2020), which is given by

$$T = \Omega \left( \left( \frac{L^4}{(1-\gamma)^6 \epsilon^2} \ln \frac{DL^4}{(1-\gamma)^7 \epsilon^2} \right)^{\frac{1}{\omega}} + \left( \frac{L^2}{1-\gamma} \ln \frac{1}{(1-\gamma)^2 \epsilon} \right)^{\frac{1}{1-\omega}} \right), \tag{17}$$

where $\omega \in (1/3, 1)$. Our result in Corollary 2 improves this characterization by orders of magnitude in terms of the dependence of the time complexity on all key parameters $\epsilon, D, 1 - \gamma, L$ (see Table 1). In particular, the dependence on the covering number $L$ in (17) is optimized by choosing $\omega = 2/3$, under which Corollary 2 improves (17) by a factor of at least $\mathcal{O}(L^5)$. Comparing our results to the lower bound obtained in Azar et al. (2013), the order of $\varepsilon$ is tight up to a logarithm factor.

## 4.2 Proof Idea of Theorem 2

Differently from the proof of Theorem 1 that does per iteration analysis, the central idea to analyze the asynchronous case is to capture the learning error in terms of the key noise and error terms over all the preceding iterations (see equation (36) in Appendix F). A novel development lies in the new method for analyzing the noise and error terms that involve the Bernoulli switching parameters specifically in double Q-learning. For example, we construct an augmented Markov chain to the underlying MDP to derive a concentration inequality on the visitation probability of an $(s, a)$-pair in a Q-table when subject to both Markovian sampling and Bernoulli switching (see Lemma 5). Moreover, we develop a conditional concentration analysis when bounding certain learning error term in the presence of Bernoulli switching (see Appendix G). These new techniques play a critical role in improving the complexity bound in terms of its dependence on the sampling related parameters such as $L$ in (15) or $t_{\mathrm{mix}}$ in (16). We refer the readers to Appendix F for the complete proof.

## 5 Discussion and Conclusion

In this paper, we derived finite-time bounds for double Q-learning using constant learning rates under both synchronous sampling and Makovian asynchronous sampling, which order-wisely improves the dependence on all the important factors compared to the existing results. To achieve this, we developed a systematic approach and a series of novel techniques to bound two nested stochastic approximation recursions.

Although the presented characterization does not achieve the same convergence rate as the best available results of the vanilla Q learning (which are $\tilde{\Omega}(\frac{D}{(1-\gamma)^4 \epsilon^2})$ for synchronous Q-learning and $\tilde{\Omega}(\frac{L}{(1-\gamma)^5 \epsilon^2})$ for asynchronous Q-learning so far, in terms of sample complexity. See Li et al. (2021, 2020); Chen et al. (2021a) etc., and comparison tables therein), our analysis serves as an important step toward a fully understanding of the fast convergence of double-Q learning. The current analysis in this paper treats deterministic and random rewards in the same. However, since double Q-learning essentially reduces the estimation bias caused by the randomness or errors in rewards, further improvement on the order of the dependence on the discount factor might be achieved by explicitly taking into consideration how the variance of the random reward affects the convergence rate. This requires a new and refined analysis. Moreover, a better convergence rate might also be achieved by introducing other techniques into the algorithm, such as mini-batches. Experimentally, there is no conclusive comparison between Q-learning and Double Q-learning in the literature. For example, a recent experimental study (Weng et al., 2020b) under a similar setting to this paper shows that when using the same step size, Q-learning has a faster rate of convergence initially but suffers from a higher mean-squared error. On the other hand, when double Q-learning uses twice the step size of Q-learning, their simulation shows that double Q-learning achieves a faster initial convergence rate, at the cost of a possibly worse mean-squared error than Q-learning. This topic is still up to the future investigation to have a solid conclusion.

Another important yet challenging topic is the convergence guarantee for double Q-learning with function approximation. In addition to the lack of the contraction property of the Bellman operator in the function approximation setting, it is likely that neither of the two Q-estimators converges, or they do not converge to the same point even if they both converge. Characterizing the conditions under which double Q-learning with function approximation converges is still an open problem.

## Acknowledgements

The work of L. Zhao was supported in part by the Singapore Ministry of Education Academic Research Fund Tier 1 under the grant R-263-000-E60-133. The work of H. Xiong and Y. Liang was supported in part by the U.S. National Science Foundation under the grant CCF-1761506.

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
