# Supplementary Materials

## A    Proof of Nested SA Formulation and its Properties

**Derivation of the $r_t$-recursion:** We let

$$\tilde{\alpha}_t(s,a) = \begin{cases} \alpha_t\beta_t, & \text{for synchronous version,} \\ \alpha_t\beta_t\tau_t(s,a), & \text{for asynchronous version.} \end{cases}$$

Then for both (2) and (3), we can write the error dynamics as

$$\begin{aligned}
r_{t+1}(s,a) &= (1 - \tilde{\alpha}_t(s,a))r_t(s,a) + \tilde{\alpha}_t(s,a)\left(R_t(s,a,s') + \gamma Q_t^B(s',a^*) - Q^*(s,a)\right) \\
&= (1 - \tilde{\alpha}_t(s,a))r_t(s,a) + \tilde{\alpha}_t(s,a)\left(R_t(s,a,s') + \gamma Q_t^A(s',a^*) - \widehat{\mathcal{T}}_t Q^*(s,a)\right. \\
&\quad \left. + \widehat{\mathcal{T}}_t Q^*(s,a) - Q^*(s,a) + \gamma Q_t^B(s',a^*) - \gamma Q_t^A(s',a^*)\right) \\
&= (1 - \tilde{\alpha}_t(s,a))r_t(s,a) + \tilde{\alpha}_t(s,a)\left(\widehat{\mathcal{T}}_t Q_t^A(s,a) - \widehat{\mathcal{T}}_t Q^*(s,a)\right. \\
&\quad \left. + \widehat{\mathcal{T}}_t Q^*(s,a) - Q^*(s,a) + \gamma Q_t^B(s',a^*) - \gamma Q_t^A(s',a^*)\right) \\
&:= (1 - \tilde{\alpha}_t(s,a))r_t(s,a) + \tilde{\alpha}_t(s,a)\left(\widehat{\mathcal{T}}_t Q_t^A(s,a) - \widehat{\mathcal{T}}_t Q^*(s,a)\right. \\
&\quad \left. + \varepsilon_t(s,a) + \gamma Q_t^B(s',a^*) - \gamma Q_t^A(s',a^*)\right),
\end{aligned}$$

which is exactly (4).

**Uniform bound of $\varepsilon_t$:** It follows from the definition that

$$\begin{aligned}
|\varepsilon_t(s,a)| &= \left|\widehat{\mathcal{T}}_t Q^*(s,a) - \mathcal{T} Q^*(s,a)\right| \\
&= \left|R_t(s,a,s') + \gamma\max_{a'\in\mathcal{A}}Q^*(s',a') - R_{sa}^{s'} - \gamma\mathbb{E}_{s'}\max_{a'\in\mathcal{A}}Q^*(s',a')\right| \\
&\le 1 + \gamma\left(\max_{(s,a)\in\mathcal{S}\times\mathcal{A}}Q^*(s,a) - \min_{(s,a)\in\mathcal{S}\times\mathcal{A}}Q^*(s,a)\right) \\
&= 1 + \frac{\gamma}{1-\gamma} \\
&= \frac{1}{1-\gamma}.
\end{aligned}$$

For brevity, we denote $\|\varepsilon_t\| \le V_{\max} := \frac{1}{1-\gamma}$.

**Derivation of the $\nu_t$-recursion:** We let

$$\hat{\alpha}_t(s,a) = \begin{cases} \alpha, & \text{for synchronous version,} \\ \alpha\tau_t(s,a), & \text{for asynchronous version.} \end{cases}$$

Then, based on (2) or (3), we have $\forall t \ge 1$,

$$\begin{aligned}
\nu_{t+1}(s,a) &= Q_{t+1}^B(s,a) - Q_{t+1}^A(s,a) \\
&= (1 - \hat{\alpha}_t(s,a)(1-\beta_t))\,Q_t^B(s,a) + \hat{\alpha}_t(s,a)(1-\beta_t)\left(R_t(s,a,s') + \gamma Q_t^A(s',b^*)\right) \\
&\quad - (1 - \hat{\alpha}_t(s,a)\beta_t)Q_t^A(s,a) - \hat{\alpha}_t(s,a)\beta_t\left(R_t(s,a,s') + \gamma Q_t^B(s',a^*)\right) \\
&= (1 - \hat{\alpha}_t(s,a))\nu_t(s,a) + \hat{\alpha}_t(s,a)\left[(1-\beta_t)\left(R_t(s,a,s') + \gamma Q_t^A(s',b^*) - Q_t^A(s,a)\right)\right. \\
&\quad \left. + \beta_t\left(Q_t^B(s,a) - R_t(s,a,s') - \gamma Q_t^B(s',a^*)\right)\right] \\
&\overset{(i)}{=} (1 - \hat{\alpha}_t(s,a))\nu_t(s,a) + \hat{\alpha}_t(s,a)H_t(s,a), \tag{18}
\end{aligned}$$

where (i) follows from the definition

$$H_t = (1-\beta_t)\left(R_t(s,a,s') + \gamma Q_t^A(s',b^*) - Q_t^A(s,a)\right) + \beta_t\left(Q_t^B(s,a) - R_t(s,a,s') - \gamma Q_t^B(s',a^*)\right).$$

We further define $\mathcal{H}_t = \mathbb{E}(H_t|\mathcal{F}_t)$ and $\mu_t = H_t - \mathcal{H}_t$. Then (6) readily follows from (18).

**Quasi-contractive Property of** $\mathcal{H}_t(\nu_t)$ (see also Xiong et al. (2020)): By direct calculation using the definition of $H_t$ and $\mathcal{F}_t$, we have

$$\mathbb{E}(H_t(s,a)|\mathcal{F}_t) = \frac{1}{2}\nu_t(s,a) + \frac{\gamma}{2}\mathbb{E}_{s'}\left(Q_t^B(s',a^*) - Q_t^A(s',b^*)\right), \qquad (19)$$

where we used the fact that $\mathbb{E}(\beta_t) = 0.5$. It follows from (19) that

$$
\begin{aligned}
|\mathbb{E}(H_t(s,a)|\mathcal{F}_t)| &\leq \frac{1}{2}|\nu_t(s,a)| + \frac{\gamma}{2}\mathbb{E}_{s'}\left|Q_t^B(s',a^*) - Q_t^A(s',b^*)\right| \\
&\leq \frac{1}{2}\|\nu_t\| + \frac{\gamma}{2}\mathbb{E}_{s'}\begin{cases} Q_t^B(s',a^*) - Q_t^A(s',b^*) & \text{if } Q_t^B(s',a^*) \geq Q_t^A(s',b^*) \\ Q_t^A(s',b^*) - Q_t^B(s',a^*) & \text{if } Q_t^B(s',a^*) < Q_t^A(s',b^*) \end{cases} \\
&\leq \frac{1}{2}\|\nu_t\| + \frac{\gamma}{2}\mathbb{E}_{s'}\begin{cases} Q_t^B(s',b^*) - Q_t^A(s',b^*) & \text{if } Q_t^B(s',a^*) \geq Q_t^A(s',b^*) \\ Q_t^A(s',a^*) - Q_t^B(s',a^*) & \text{if } Q_t^B(s',a^*) < Q_t^A(s',b^*) \end{cases} \\
&\leq \frac{1+\gamma}{2}\|\nu_t\|,
\end{aligned}
$$

which implies that

$$\|\mathbb{E}(H_t|\mathcal{F}_t)\| \leq \frac{1+\gamma}{2}\|\nu_t\|.$$

**Boundedness of** $\mu_t$: By the definition of $H_t$ in (18) and $\mathcal{H}_t$ in (19), we have if $\beta_t = 0$,

$$
\begin{aligned}
|\mu_t(s,a)| &= \left|R_t(s,a,s') + \gamma Q_t^A(s',b^*) - Q_t^A(s,a) - \frac{1}{2}\nu_t(s,a) + \frac{\gamma}{2}\mathbb{E}_{s'}\left(Q_t^B(s',a^*) - Q_t^A(s',b^*)\right)\right| \\
&\leq |R_t(s,a,s')| + \gamma\left|Q_t^A(s',b^*)\right| + \frac{1}{2}\left(\left|Q_t^A(s,a)\right| + \left|Q_t^B(s,a)\right|\right) + \frac{\gamma}{2}\left(\left|Q_t^B(s',a^*) - Q_t^A(s',b^*)\right|\right) \\
&\leq 1 + \frac{\gamma}{1-\gamma} + \frac{1}{1-\gamma} + \frac{\gamma}{2(1-\gamma)} \\
&= 3V_{\max}.
\end{aligned}
$$

The case of $\beta_t = 1$ follows similarly and we omit the detailed proof here. Therefore, we conclude that $|\mu_t(s,a)| \leq 3V_{\max}$.

## B  Proof of Theorem 1

We first give a high-level idea about the difference between our approach and that in Xiong et al. (2020). Our central goal here is to bound the convergence error via a pair of nested SA recursions, where the outer SA captures the learning error dynamics between one Q-estimator and the global optimum, and the inner SA captures the error propagation between the two Q-estimators. Xiong et al. (2020) constructed two block-wisely decreasing bounds for the nested SAs and characterized only a block-wise convergence. Such an approach is rather complicated, and does not appear to extend easily to the constant learning rate case. In contrast, we take a very different yet more direct approach here. We devise new analysis techniques to directly bound both the inner and outer error dynamics per iteration. We then treat the output of the inner SA as a noise term in the outer SA, and combine the two bounds to establish the finite-time error bound of the learning error.

The main proof consists of five steps. The main proof utilizes a few propositions, the proofs of which are provided in the next a few sections.

**Step I: Deriving a template finite-time bound applicable to both SAs.**

Consider the following general SA algorithm with the unique fixed point $\theta^* = 0$:

$$\theta_{t+1} = (1-\alpha_t)\theta_t + \alpha_t\left(\mathcal{G}_t(\theta_t) + \varepsilon_t + \gamma\nu_t\right), \qquad (20)$$

for all $t \geq 1$, where $\theta_t \in \mathbb{R}^n$ and $\alpha_t \in [0,1)$ is a general time-varying learning rate. Note that (20) includes both (4) and (6) as special cases.

We bound $\theta_t$ in the following proposition. Note that differently from Wainwright (2019b), our analysis below only requires the quasi-contractive property and the fact that $\alpha_t \in [0,1)$. Moreover, we treat

the noise terms $\varepsilon_t$ and $\nu_t$ separately rather than combining them together into the $W_t$-recursion. This is because for double Q-learning, the noise term $\nu_t$ has its own dynamics which is significantly more complex than the noise $\varepsilon_t$. Treating them together as one noise term does not yield sharp bounds.

**Proposition 1.** *Consider an SA given in (20). Suppose $\mathcal{G}_t$ is quasi-contractive with a constant parameter $\gamma \in (0, 1)$, that is, $\|\mathcal{G}_t(\theta_t)\| \leq \gamma \|\theta_t\|$. Then for any learning rate $\alpha_t \in [0, 1)$, the iterates $\{\theta_t\}$ satisfy*

$$\|\theta_t\| \leq \prod_{k=1}^{t-1}(1-(1-\gamma)\alpha_k)\|\theta_1\|+\gamma\alpha_{t-1}\left(\|W_{t-1}\|+\|\nu_{t-1}\|\right)$$
$$+\gamma\sum_{k=1}^{t-2}\left\{\prod_{l=k+1}^{t-1}(1-(1-\gamma)\alpha_l)\right\}\alpha_k\left(\|W_k\|+\|\nu_k\|\right)+\|W_t\|, \tag{21}$$

*where the sequence $\{W_t\}$ is given by $W_{t+1} = (1 - \alpha_t)W_t + \alpha_t\varepsilon_t$ with $W_1 = \mathbf{0}$.*

*Proof.* See Appendix C. $\qquad\square$

Note that the SA recursion (6) is a special case of (20) by setting $\nu_t = \mathbf{0}$. Therefore, Proposition 1 is readily applicable to both (4) and (6).

**Step II: Bounding outer SA dynamics $\mathbb{E}\|r_t\|$ by inner SA dynamics $\mathbb{E}\|\nu_t\|$.**

We apply Proposition 1 to the error dynamics (4) of $r_t$. Recall that $\mathcal{G}_t$ in (4) is quasi-contractive, which satisfies $\|\mathcal{G}_t(r_t)\| \leq \gamma \|r_t\|$. Now construct the following recursion:

$$W_{t+1} = (1 - \tilde{\alpha}_t)W_t + \tilde{\alpha}_t\varepsilon_t, \quad \text{with initialization } W_1 = \mathbf{0}. \tag{22}$$

Further define $\tilde{\nu}_t(s, a) = \nu_t(s', a^*)$ and note that $\|\tilde{\nu}_t\| \leq \|\nu_t\|$ because all the elements of $\tilde{\nu}_t$ come from $\nu_t$. Then applying Proposition 1 to the SA (4) yields

$$\|r_t\| \leq \prod_{k=1}^{t-1}(1-(1-\gamma)\tilde{\alpha}_k)\|r_1\| + \gamma\tilde{\alpha}_{t-1}\left(\|W_{t-1}\| + \|\tilde{\nu}_{t-1}\|\right)$$
$$+\gamma\sum_{k=1}^{t-2}\left\{\prod_{l=k+1}^{t-1}(1-(1-\gamma)\tilde{\alpha}_l)\right\}\tilde{\alpha}_k\left(\|W_k\| + \|\tilde{\nu}_k\|\right) + \|W_t\|.$$

Further taking the expectation on both sides of the above bound, and denoting $h := 1 - \frac{1-\gamma}{2}\alpha$, we have

$$\mathbb{E}\|r_t\| \overset{(i)}{\leq} h^{t-1}\|r_1\| + \frac{\gamma}{2}\alpha\sum_{k=1}^{t-1}h^{t-k-1}\left(\mathbb{E}\|W_k\| + \mathbb{E}\|\tilde{\nu}_k\|\right) + \mathbb{E}\|W_t\|$$
$$\overset{(ii)}{\leq} h^{t-1}\|r_1\| + \frac{\gamma}{2}\alpha\sum_{k=1}^{t-1}h^{t-k-1}\left(\mathbb{E}\|W_k\| + \mathbb{E}\|\nu_k\|\right) + \mathbb{E}\|W_t\|, \tag{23}$$

where (i) follows because $\{\tilde{\alpha}_t\}_{t\geq 1}$ is a sequence of independent random variables, $\{\tilde{\alpha}_s\}_{s\geq t}$ is independent of $W_t$ and $\tilde{\nu}_t$, and $\mathbb{E}\tilde{\alpha}_t = \frac{\alpha_t}{2}$, and (ii) follows because $\|\tilde{\nu}_t\| \leq \|\nu_t\|$.

The bound in (23) captures the coupling between the nested SAs of double Q-learning, where $\nu_t$ of the inner SA enters as a tracking error term into the bound on $r_t$ of the outer SA. To further bound the outer SA error, we need to handle both $\|\nu_t\|$ and $\|W_t\|$, which is given in the next two steps.

**Step III: Bounding $\mathbb{E}\|W_t\|$.**

We provide the bound on the expectation of the sup-norm of $W_{t+1}$ in the following Proposition. Recall $D = |\mathcal{S}||\mathcal{A}|$ denotes the dimension of the state-action space.

**Proposition 2.** *Consider the sequence $\{W_{t+1}\}_{t\geq 1}$ generated by the recursion (22). We have*

$$\mathbb{E}\|W_{t+1}\| \leq 2\tilde{D}V_{\max}\sqrt{\alpha}, \tag{24}$$

*where $\tilde{D} = 2\sqrt{\ln 2D} + \sqrt{\pi}$ and $V_{\max}$ is defined in (5).*

*Proof.* See Appendix D for the complete proof. Here we briefly describe the key idea of the proof, which lies in the construction of an $\mathcal{F}_t$-martingale sequence $\{\tilde{W}_i\}_{1 \leq i \leq t+1}$ from the recursion (22), where $\tilde{W}_{t+1} = W_{t+1}$ and $\tilde{W}_1 = \mathbf{0}$. Based on the uniform bound of $W_t$ proved in Lemma 1, we are able to bound the difference sequence $\left| \tilde{W}_{i+1} - \tilde{W}_i \right|$ by a geometric series (see Lemma 2). Then we apply the Azuma-Hoeffding inequality (see Lemma 3) to $\{\tilde{W}_i\}_{1 \leq i \leq t+1}$ and further use Lemma 4 to obtain the claimed bound. $\qquad\square$

**Step IV: Bounding inner SA dynamics $\mathbb{E} \left\| \nu_t \right\|$.**

Now our goal is to bound $\mathbb{E} \left\| \nu_t \right\|$. The inner SA captures the difference between two Q-estimators, and is an error existing specifically in double Q-learning. Bounding $\mathbb{E} \left\| \nu_t \right\|$ is one of the key steps to handle the coupling between two nested SAs.

Recall that in the $\nu_t$-recursion (6), the operator $\mathcal{H}_t$ is quasi-contractive, which satisfies $\left\| \mathcal{H}_t(\nu_t) \right\| \leq \frac{1+\gamma}{2} \left\| \nu_t \right\|$. Then by constructing the following recursion:

$$M_{t+1} = (1 - \alpha)M_t + \alpha\mu_t, \quad \text{with initialization } M_1 = \mathbf{0}, \tag{25}$$

we apply Proposition 1 and take the expectations on both sides to have

$$\mathbb{E} \left\| \nu_t \right\| \leq \tilde{\gamma}\alpha \sum_{k=1}^{t-1} h^{t-k-1} \mathbb{E} \left\| M_k \right\| + \mathbb{E} \left\| M_t \right\|, \tag{26}$$

where $\tilde{\gamma} := \frac{1+\gamma}{2}$ is the quasi-contractive coefficient of $\mathcal{H}_t$ and recall that with loss of generality we assumed $\nu_1 = \mathbf{0}$.

We further provide the bound on $\mathbb{E} \left\| M_t \right\|$ in the following proposition.

**Proposition 3.** *Consider the sequence $\{M_{t+1}\}_{t \geq 1}$ generated by the recursion (25). We have*

$$\mathbb{E} \left\| M_{t+1} \right\| \leq 2\tilde{D}V_{\max}\sqrt{2\alpha},$$

*where $\tilde{D} := 2\sqrt{\ln 2D} + \sqrt{\pi}$.*

*Proof.* See Appendix E. $\qquad\square$

Taking the expectation on both sides of (26) and using Proposition 3, we obtain

$$\mathbb{E} \left\| \nu_t \right\| \leq \tilde{\gamma}\alpha \sum_{k=2}^{t-1} h^{t-k-1} \mathbb{E} \left\| M_k \right\| + \mathbb{E} \left\| M_t \right\|$$

$$\leq \frac{4}{1-\gamma} \tilde{D}V_{\max}\sqrt{2\alpha}.$$

The above bound indicates that the inner SA error can be controlled by the learning rate, which thus plays an important role to guarantee the overall convergence by ensuring that such an error asymptotically vanishes when entering into the outer SA error.

**Step V: Deriving overall finite-time complexity.**

Substituting the bounds on $\mathbb{E} \left\| W_t \right\|$ and $\mathbb{E} \left\| \nu_t \right\|$ into (23), we have $\forall t \geq 3$,

$$\mathbb{E} \left\| r_{t+1} \right\| \leq h^{t-1} \left\| r_1 \right\| + \frac{\gamma}{2}\alpha \sum_{k=1}^{t-1} h^{t-k-1} \left( \mathbb{E} \left\| W_k \right\| + \mathbb{E} \left\| \nu_k \right\| \right) + \mathbb{E} \left\| W_t \right\|$$

$$\leq h^{t-1} \left\| r_1 \right\| + \frac{2\tilde{D}}{(1-\gamma)^2}\sqrt{\alpha} + \frac{4\tilde{D}}{(1-\gamma)^3}\sqrt{2\alpha}$$

$$\leq h^{t-1} \left\| r_1 \right\| + \frac{6\tilde{D}}{(1-\gamma)^3}\sqrt{2\alpha},$$

where $\tilde{D} = 2\sqrt{\ln 2D} + \sqrt{\pi}$. The above expectation bound is equivalent to the high probability bound on $\left\| r_t \right\|$ by replacing $\ln 2D$ with $c\ln \frac{2D}{\delta}$ for some universal constant $c > 0$ (see for example Wainwright (2019b)). We readily have that (8) holds with the probability of at least $1 - \delta$. This completes the proof. $\qquad\square$

## C    Proof of Proposition 1

To proceed the proof, we first construct the following useful recursions:

$$W_{t+1} = (1 - \alpha_t)W_t + \alpha_t \varepsilon_t, \quad \text{with initialization } W_1 = \mathbf{0},$$
$$b_{t+1} = (1 - (1-\gamma)\alpha_t) b_t, \quad \text{with initialization } b_1 = \|\theta_1\| \mathbf{1},$$
$$g_{t+1} = (1 - (1-\gamma)\alpha_t) g_t + \gamma\alpha_t (\|W_t\| + \|\nu_t\|) \mathbf{1}, \quad \text{with initialization } g_1 = \mathbf{0},$$

where $\mathbf{1}$ denotes all-ones vector and $\mathbf{0}$ denotes all-zeros vector with appropriate dimensions. Note that $\{b_t\}_{t\geq 1}$ and $\{g_t\}_{t\geq 1}$ are both non-negative sequences satisfying $b_t = \|b_t\| \mathbf{1}$ and $g_t = \|g_t\| \mathbf{1}$ for all $t \geq 1$.

Then we have the following sandwich bound on $\theta_t$ given by

$$-b_t - g_t + W_t \preceq \theta_t \preceq b_t + g_t + W_t, \tag{27}$$

where $\preceq$ denotes the elementwise $\leq$ relationship.

We next prove (27) by induction. For $t = 1$, we have $-b_1 \preceq \theta_1 \preceq b_1$, which holds easily since $b_1 = \|\theta_1\| \mathbf{1}$. Now suppose (27) holds for some $t \geq 1$, and we prove it holds for $t + 1$.

We first note that

$$\|\theta_t\| \mathbf{1} \preceq \max\{\|b_t + g_t + W_t\| \mathbf{1}, \|-b_t - g_t + W_t\| \mathbf{1}\}$$
$$\preceq b_t + g_t + \|W_t\| \mathbf{1}, \tag{28}$$

since $x_t = \|x_t\| \mathbf{1}$ for $x \in \{b, g\}$.

For the upper bound, we have

$$\theta_{t+1} = (1 - \alpha_t)\theta_t + \alpha_t (\mathcal{G}_t(\theta_t) + \varepsilon_t + \gamma\nu_t)$$

$$\overset{(i)}{\preceq} (1 - \alpha_t)(b_t + g_t + W_t) + \alpha_t (\gamma \|\theta_t\| \mathbf{1} + \varepsilon_t + \gamma \|\nu_t\| \mathbf{1})$$

$$\overset{(ii)}{\preceq} (1 - \alpha_t)(b_t + g_t + W_t) + \alpha_t [\gamma (b_t + g_t + \|W_t\| \mathbf{1}) + \varepsilon_t + \gamma \|\nu_t\| \mathbf{1}]$$

$$= \underbrace{(1 - (1-\gamma)\alpha_t)b_t}_{b_{t+1}} + \underbrace{(1 - (1-\gamma)\alpha_t)g_t + \gamma\alpha_t (\|W_t\| + \|\nu_t\|) \mathbf{1}}_{g_{t+1}}$$

$$+ \underbrace{(1 - \alpha_t)W_t + \alpha_t \varepsilon_t}_{W_{t+1}},$$

where (i) follows from the induction assumption and the quasi-contractive property of $\mathcal{G}_t$, and (ii) follows from (28).

For the lower bound, we have

$$\theta_{t+1} = (1 - \alpha_t)\theta_t + \alpha_t (\mathcal{G}_t(\theta_t) + \varepsilon_t + \gamma\nu_t)$$

$$\overset{(i)}{\succeq} (1 - \alpha_t)(-b_t - g_t + W_t) + \alpha_t (-\gamma \|\theta_t\| \mathbf{1} + \varepsilon_t - \gamma \|\nu_t\| \mathbf{1})$$

$$\overset{(ii)}{\succeq} (1 - \alpha_t)(-b_t - g_t + W_t) + \alpha_t [-\gamma (b_t + g_t + \|W_t\| \mathbf{1}) + \varepsilon_t - \gamma \|\nu_t\| \mathbf{1}]$$

$$= \underbrace{-(1 - (1-\gamma)\alpha_t)b_t}_{-b_{t+1}} + \underbrace{-[(1 - (1-\gamma)\alpha_t)g_t + \gamma\alpha_t (\|W_t\| + \|\nu_t\|) \mathbf{1}]}_{-g_{t+1}}$$

$$+ \underbrace{(1 - \alpha_t)W_t + \alpha_t \varepsilon_t}_{W_{t+1}},$$

where (i) follows from the induction assumption and the quasi-contractive property of $\mathcal{G}_t$, and (ii) follows from (28).

Thus we have proven that (27) holds for $t + 1$. By induction, it holds for all $t \geq 1$. Finally, we have

$$\|\theta_t\| \leq \prod_{k=1}^{t-1} (1 - (1-\gamma)\alpha_k) \|\theta_1\| + \gamma\alpha_{t-1} (\|W_{t-1}\| + \|\nu_{t-1}\|)$$

$$+ \gamma \sum_{k=1}^{t-2} \left\{ \prod_{l=k+1}^{t-1} (1 - (1-\gamma)\alpha_l) \right\} \alpha_k (\|W_k\| + \|\nu_k\|) + \|W_t\|,$$

where the first term on the right hand side is $\|b_t\|$ and the sum of the next two terms correspond to $\|g_t\|$. $\square$

## D  Proof of Proposition 2

We first prove some useful lemmas.

**Lemma 1.** *Consider the sequence $\{W_{t+1}\}_{t \geq 1}$ generated by the recursion (22). Then we have $|W_{t+1}| \leq V_{\max}$.*

*Proof.* We prove it by induction. For $t = 1$, we have

$$|W_2| = |\tilde{\alpha}_1 \varepsilon_1| \leq \alpha V_{\max} < V_{\max}.$$

Suppose $|W_t| \leq V_{\max}$ for some $t \geq 2$. Then it follows that,

$$|W_{t+1}| \leq (1 - \tilde{\alpha}_t)|W_t| + |\tilde{\alpha}_t \varepsilon_t| \leq (1 - \tilde{\alpha}_t)V_{\max} + \tilde{\alpha}_t V_{\max} = V_{\max}.$$

Thus by induction $|W_{t+1}| \leq V_{\max}$ for all $t \geq 1$. $\square$

**Lemma 2.** *Consider the martingale sequence $\{\tilde{W}_i\}_{1 \leq i \leq T+1}$, defined in (31) where $T \geq 1$. We have the corresponding difference sequence bounded by*

$$\left|\tilde{W}_{i+1} - \tilde{W}_i\right| \leq \tilde{W}_{i+1} - \tilde{W}_i = 2\alpha\left(1 - \frac{\alpha}{2}\right)^{(T-i)}V_{\max}, \quad 1 \leq i \leq T, \tag{29}$$

*where $V_{\max}$ is the uniform bound of the noise sequence $\{\varepsilon_t\}_{t \geq 1}$ defined in (5).*

*Proof.* By the definition of $\{\tilde{W}_i\}_{1 \leq i \leq T+1}$, we have

$$\tilde{W}_{i+1} - \tilde{W}_i = (1 - \frac{\alpha}{2})^{(T-i)}\alpha_i \Gamma_i, \quad 1 \leq i \leq T, \tag{30}$$

where $\Gamma_i := (\frac{1}{2} - \beta_i)W_i + \beta_i \varepsilon_i$, for all $1 \leq i \leq T$. Since $W_1 = 0$, we easily have $|\Gamma_1| \leq V_{\max}$. For $i \geq 2$, we have

$$|\Gamma_i| = \begin{cases} \left|-\frac{1}{2}W_i + \varepsilon_i\right| & \text{if } \beta_t = 1 \\ \frac{1}{2}|W_i| & \text{if } \beta_t = 0 \end{cases} \leq \frac{1}{2}|W_i| + |\varepsilon_i| \overset{(i)}{\leq} \frac{3}{2}V_{\max} < 2V_{\max},$$

where (i) follows from Lemma 1. Substituting the above bound into (30) completes the proof. $\square$

**Lemma 3.** *(Azuma-Hoeffding Inequality) Suppose $\{S_n\}_{n \geq 1}$ is a martingale such that $S_0 = 0$ and $|S_i - S_{i-1}| \leq d_i$ almost surely for some constants $d_i$, $1 \leq i \leq n$. Then, for all $t \geq 0$,*

$$\mathbb{P}\left(|S_n| \geq \rho\right) \leq 2\exp\left(-\frac{\rho^2}{2\sum_{i=1}^n d_i^2}\right).$$

The following lemma slightly extends Wainwright (2019a, Exercise 2.8 (a)) to handle the case where $b = 0$. The proof is similar and we include it here for completeness.

**Lemma 4.** *Suppose $Z$ is a non-negative random variable satisfying the concentration inequality $\mathbb{P}(Z \geq \rho) \leq C\exp\left(-\rho^2/\sigma^2\right)$, $\forall \rho > 0$, for some $C > 1, \sigma > 0$. Then we have $\mathbb{E}(Z) \leq \sigma\left(\sqrt{\ln C} + \frac{\sqrt{\pi}}{2}\right)$.*

*Proof.* By the expectation formula of non-negative random variables, we have

$$\mathbb{E}\left(Z\right) = \int_0^\infty \mathbb{P}(Z \geq \rho)d\rho \leq \int_0^\infty 1 \wedge C\exp\left(-\frac{\rho^2}{\sigma^2}\right)d\rho$$

$$= \int_0^{\sigma\sqrt{\ln C}} 1 d\rho + \int_{\sigma\sqrt{\ln C}}^\infty C\exp\left(-\frac{\rho^2}{\sigma^2}\right)d\rho$$

$$= \sigma\sqrt{\ln C} + \int_{\sigma\sqrt{\ln C}}^\infty \exp\left(-\frac{\rho^2 - \sigma^2\ln C}{\sigma^2}\right)d\rho$$

$$\overset{(i)}{\leq} \sigma\sqrt{\ln C} + \int_{\sigma\sqrt{\ln C}}^\infty \exp\left(-\frac{\left(\rho - \sigma\sqrt{\ln C}\right)^2}{\sigma^2}\right)d\rho$$

$$= \sigma\sqrt{\ln C} + \int_0^\infty \exp\left(-\frac{z^2}{\sigma^2}\right)dz$$

$$= \sigma\left(\sqrt{\ln C} + \frac{\sqrt{\pi}}{2}\right),$$

where (i) follows because $\rho^2 - a^2 \geq (\rho - a)^2$ for $\rho \geq a \geq 0$. $\qquad\square$

**Proof of Proposition 2:**

Recall the definition of $\mathcal{F}_t$ in (7), and we have

$$\mathbb{E}\left(W_{t+1}\,|\mathcal{F}_t\right) = \mathbb{E}\left((1 - \tilde{\alpha}_t)W_t + \tilde{\alpha}_t\varepsilon_t\,|\mathcal{F}_t\right)$$

$$\overset{(i)}{=} (1 - \frac{\alpha}{2})W_t + \frac{\alpha}{2}\mathbb{E}\left(\varepsilon_t\,|\mathcal{F}_t\right)$$

$$\overset{(ii)}{=} (1 - \frac{\alpha}{2})W_t + \underbrace{\frac{\alpha}{2}\mathbb{E}\left(\varepsilon_t\right)}_{=0}$$

$$= (1 - \frac{\alpha}{2})W_t,$$

where (i) follows because $\beta_t, W_t, \varepsilon_t$ are independent, $\sigma(W_t) \subset \mathcal{F}_t$ (because $W_t$ is a measurable function of $\{\beta_{k-1}, s_k\}_{2 \leq k \leq t}$), and $\mathbb{E}(\tilde{\alpha}_t) = \alpha\mathbb{E}(\beta_t) = \frac{\alpha_t}{2}$; (ii) follows because $\sigma(\varepsilon_t) = \sigma(s_{t+1})$ which is independent of $\mathcal{F}_t$. We then readily have $\mathbb{E}\left(W_{t+1}\right) = \mathbb{E}\left(W_{t+1}\,|\mathcal{F}_1\right) = \mathbf{0}$.

Therefore, if we define

$$\tilde{W}_i := (1 - \frac{\alpha}{2})^{t+1-i}W_i, \quad 1 \leq i \leq t+1, \tag{31}$$

then $\{\tilde{W}_i\}_{1 \leq i \leq t+1}$ is a martingale sequence with $\tilde{W}_{t+1} = W_{t+1}$ and $\tilde{W}_1 = \mathbf{0}$, for any $t \geq 1$.

Now using Lemma 2, we have

$$d_i := \left|\tilde{W}_{i+1} - \tilde{W}_i\right| \leq 2\alpha\left(1 - \frac{\alpha}{2}\right)^{t-i}V_{\max}, \quad 1 \leq i \leq t,$$

and thus

$$\sum_{i=1}^t d_i^2 \leq 4\alpha^2 V_{\max}^2 \sum_{i=1}^t \left(1 - \frac{\alpha}{2}\right)^{2(t-i)} \leq \frac{16\alpha V_{\max}^2}{4 - \alpha}.$$

Then using the Azuma-Hoeffding Inequality (see Lemma 3) and the union bound for the maximum norm, we have,

$$\mathbb{P}\left(\|W_{t+1}\| \geq \rho\right) = \mathbb{P}\left(\max_{(s,a)\in\mathcal{S}\times\mathcal{A}}|W_{t+1}| \geq \rho\right)$$

$$\leq D\mathbb{P}\left(|W_{t+1}| \geq \rho\right)$$

$$\leq 2D \exp\left(-\frac{\rho^2}{2\sum_{i=1}^t d_i^2}\right)$$

$$\leq 2D \exp\left(-\frac{(4-\alpha)\rho^2}{32\alpha V_{\max}^2}\right).$$

Then it follows that

$$\mathbb{E}\|W_{t+1}\| = \int_0^\infty \mathbb{P}(\|W_{t+1}\| \geq \rho)d\rho,$$

$$\leq 2D \int_0^\infty \exp\left(-\frac{(4-\alpha)\rho^2}{32\alpha V_{\max}^2}\right)d\rho$$

$$\overset{(i)}{\leq} 4V_{\max}\sqrt{\frac{2\alpha}{4-\alpha}}\left(\sqrt{\ln 2D} + \frac{\sqrt{\pi}}{2}\right)$$

$$\overset{(ii)}{=} 2\tilde{D}V_{\max}\sqrt{\frac{2\alpha}{4-\alpha}}$$

$$\overset{(ii)}{\leq} 2\tilde{D}V_{\max}\sqrt{\alpha},$$

where (i) follows from Lemma 4, (ii) follows from the definition $\tilde{D} := 2\sqrt{\ln 2D} + \sqrt{\pi}$, and (iii) follows because $\alpha < 1$.

$\square$

# E   Proof of Proposition 3

Since $\{\mu_t\}_{t\geq 1}$ is a martingale difference sequence, we have

$$\mathbb{E}\left(M_{t+1}\,|\,\mathcal{F}_t\right) = \mathbb{E}\left((1-\alpha)M_t + \alpha\mu_t\,|\,\mathcal{F}_t\right)$$
$$= (1-\alpha)M_t + \alpha\mathbb{E}\left(\mu_t\,|\,\mathcal{F}_t\right)$$
$$= (1-\alpha)M_t.$$

Therefore, if we define

$$\tilde{M}_i := (1-\alpha)^{t-i+1}M_i, \quad 1 \leq i \leq t+1,$$

then $\{\tilde{M}_i\}_{1\leq i\leq t+1}$ is a martingale sequence with $\tilde{M}_{t+1} = M_{t+1}$ and $\mathbb{E}\left(\tilde{M}_1\right) = \mathbf{0}$, for any $t \geq 1$.

By the construction of $\{\tilde{M}_i\}_{1\leq i\leq t+1}$, it can be derived that

$$\tilde{M}_{i+1} - \tilde{M}_i = (1-\alpha)^{t-i}\alpha\mu_i, \quad 1 \leq i \leq t.$$

Further using the bound $|\mu_t| \leq 3V_{\max} := \tilde{V}_{\max}$, we have

$$d_i := \left|\tilde{M}_{i+1} - \tilde{M}_i\right| \leq (1-\alpha)^{t-i}\alpha\tilde{V}_{\max}, \quad 1 \leq i \leq t,$$

and thus

$$\sum_{i=1}^t d_i^2 = \sum_{i=1}^t (1-\alpha)^{2(t-i)}\alpha^2\tilde{V}_{\max}^2 \leq \frac{\alpha}{2-\alpha}\tilde{V}_{\max}^2.$$

Now using the Azuma-Hoeffding Inequality (see Lemma 3), we have for $t \geq 1$,

$$\mathbb{P}\left(|M_{t+1}| \geq \rho\right) \leq 2\exp\left(-\frac{\rho^2}{2\sum_{i=1}^t d_i^2}\right) \leq 2\exp\left(-\frac{(2-\alpha)\rho^2}{2\alpha\tilde{V}_{\max}^2}\right). \tag{32}$$

It then follows from the union bound of the max operator that

$$\mathbb{P}\left(\|M_{t+1}\| \geq \rho\right) = \mathbb{P}\left(\max_{(s,a)\in\mathcal{S}\times\mathcal{A}}|M_{t+1}| \geq \rho\right) \leq D\mathbb{P}\left(|M_{t+1}| \geq \rho\right).$$

Furthermore, we have

$$
\begin{aligned}
\mathbb{E}\left\|M_{t+1}\right\| &= \int_0^\infty \mathbb{P}(\|M_{t+1}\| \geq \rho) d\rho \\
&\leq 2D \int_0^\infty \exp\left(-\frac{(2-\alpha)\rho^2}{2\alpha \tilde{V}_{\max}^2}\right) d\rho \\
&\stackrel{\text{(i)}}{\leq} \tilde{V}_{\max} \sqrt{\frac{2\alpha}{2-\alpha}} \left(\sqrt{\ln 2D} + \frac{\sqrt{\pi}}{2}\right) \\
&\stackrel{\text{(ii)}}{=} 2\tilde{D}V_{\max}\sqrt{\frac{2\alpha}{2-\alpha}} \\
&\stackrel{\text{(iii)}}{\leq} 2\tilde{D}V_{\max}\sqrt{\alpha},
\end{aligned}
$$

where (i) follows from Lemma 4, (ii) follows from the definition $\tilde{D} := 2\sqrt{\ln 2D} + \sqrt{\pi}$, and (iii) follows from the assumption that $\alpha < 1$. $\qquad\square$

## F   Proof of Theorem 2

We first prove some useful lemmas. The following lemma captures how often an $(s, a)$-pair is updated. Recall that $\mu_\pi$ is the stationary distribution of the underlying Markov decision process under the behavior policy $\pi$. Specifically, the lemma gives an probabilistic characterization of the number of updates for an arbitrary $(s, a)$-pair of either Q-table after a sufficient number of iterations.

**Lemma 5.** *Let $\beta_t, \tau_t(s, a)$ be defined in Section 2. Suppose Assumption 1 holds. Fix any $\delta \in (0, 1)$ and $T \geq t > \frac{886 t_{\mathrm{mix}}}{\mu_{\min}} \ln\left(\frac{4DT}{\delta}\right) := t_{\mathrm{frame}}$. Then*

$$
\forall (s_1, a_1), \qquad \mathbb{P}_{(s_1, a_1)}\left(\exists (s, a) : \sum_{i=1}^t \beta_t \tau_t(s, a) \leq \frac{1}{2} t \mu_\pi(s, a)\right) \leq \delta. \tag{33}
$$

*Proof.* The main idea of the proof lies in the construction of an auxiliary Markov chain which has the same mixing time as the original MDP under the behavior policy but only has half of its $\mu_{\min}$. The construction is inspired by the following intuition. Since $\{\beta_i\}$ is a Bernoulli random variable with expectation $\frac{1}{2}$, intuitively, double-Q learning should take two times of the iterations needed by vanilla Q-learning to visit all the states of $Q^A$ with the same high probability. To show this formally, we construct an auxiliary Markov chain by augmenting the states with $\beta_t$, namely, $\bar{M} := \{\bar{X}_t\}_{t \geq 1} = \{s_t, a_t, \beta_t\}_{\forall t \geq 1}$ with state space $\bar{\mathcal{X}} := \mathcal{S} \times \mathcal{A} \times \mathcal{B}$, where $\mathcal{B} = \{0, 1\}$. It is easy to see that such an auxiliary Markov chain is aperiodic and irreducible (and thus uniformly ergodic) given that the original Markov chain $M_o := \{X_t\}_{t \geq 1} = \{s_t, a_t\}_{t \geq 1}$ is aperiodic and irreducible. The transition probability can be calculated by

$$
\mathbb{P}\left(\bar{X}_{t+1} | \bar{X}_t\right) \stackrel{\text{(i)}}{=} \mathbb{P}\left(\beta_t\right) \mathbb{P}\left(s_{t+1}, a_{t+1} | s_t, a_t\right) \stackrel{\text{(ii)}}{=} \frac{1}{2} \pi(a_{t+1} | s_{t+1}) \mathbb{P}\left(s_{t+1} | s_t, a_t\right), \tag{34}
$$

where (i) follows from the fact that $\{\beta_t\}_{t \geq 1}$ are i.i.d Bernoulli random variables which are independent of $\{s_t, a_t\}_{t \geq 1}$, and in (ii) we denote by $\pi$ the underlying behavior policy of the Markov chain following which we take samples. Let $\bar{P}$ denote the transition probability matrix of $\bar{M}$ where the $((s, a, \beta), (s', a', \beta'))$th entry of $\bar{P}$ is $\frac{1}{2} \pi(a' | s') \mathbb{P}\left(s' | s, a\right)$. For the ease of discussion, assume that the top left $|\mathcal{S}||A| \times |\mathcal{S}||A|$ submatrix of $\bar{P}$ corresponds to the transitions between $(s, a, 1)$'s. Furthermore, let $\bar{\mu} \in \Delta(\mathcal{S} \times \mathcal{A} \times \mathcal{B})$ denote the stationary distribution of $\bar{M}$.

Let $P_o$ denote the transition probability matrix of $M_o$ where the $((s, a), (s', a'))$-th entry of $P_o$ is $\pi(a' | s') \mathbb{P}\left(s' | s, a\right)$. Let $\mu \in \Delta(\mathcal{S} \times \mathcal{A})$ be the stationary distribution of $M_o$, and thus we have $\mu P_o = \mu$, assuming that $\mu$ is a row vector. Let $P^t(\cdot | x)$ denote the distribution of $X_t$ (assuming a row vector). Then conditioned on $X_1 = x \in \mathcal{X}$, we have $P^t(\cdot | x) P_o = P^{t+1}(\cdot | x)$. By (34), we have for $\bar{M}$ that

$$
\bar{P} = \begin{bmatrix} 1 & 1 \\ 1 & 1 \end{bmatrix} \otimes \frac{1}{2} P_o, \tag{35}
$$

where $\otimes$ denotes the Kronecker product. Similarly, we call $\bar{P}^t(\cdot|\bar{x})$ the distribution of $\bar{X}_t$, conditioned on $\bar{X}_1 = \bar{x}$. It is easy to verify (using (35)) that $\bar{P}^t(\cdot|\bar{x}) = [\frac{1}{2}P^t(\cdot|x), \frac{1}{2}P^t(\cdot|x)]$ with either $\bar{x} = (x, 0)$ or $\bar{x} = (x, 1)$. Let $t \to \infty$, and we have the stationary distribution of $\bar{M}$ as $\bar{\mu} = [\frac{1}{2}\mu, \frac{1}{2}\mu]$. It follows that $\bar{\mu}_{\min} = \frac{1}{2}\mu_{\min}$.

We claim that the mixing times for $\bar{M}$ and $M_o$ are the same. To see this, we calculate the variation distances $\forall x \in \mathcal{X}$,

$$d_{\text{TV}}(P^t(\cdot|x), \mu) = \frac{1}{2}\sum_{y \in \mathcal{X}}\left|P^t(y|x) - \mu(y)\right|,$$

$$d_{\text{TV}}(\bar{P}^t(\cdot|\bar{x}), \bar{\mu}) = d_{\text{TV}}([\frac{1}{2}P^t(\cdot|x), \frac{1}{2}P^t(\cdot|x)], [\frac{1}{2}\mu, \frac{1}{2}\mu]) = \frac{1}{2}\sum_{y \in \mathcal{X}}\left|P^t(y|x) - \mu(y)\right|,$$

which are the same. Therefore we conclude that by the definition of the mixing time, the Markov chain related parameter $\mu_{\min}$ is only half of that in the Q-learning case.

Finally, applying Lemma 5 of Li et al. (2020) to the auxiliary Markov chain, which has $\bar{\mu}_{\min} = \frac{1}{2}\mu_{\min}$ and the same $t_{\text{mix}}$ as double Q-learning, we obtain (33). □

The next lemma provides a property to analyze the learning rates and the corresponding randomness.

**Lemma 6.** *Let $\tilde{\alpha}_t(s, a) = \alpha\beta_t\tau_t(s, a)$ be defined in Section 3.1. Then,*

$$\sum_{i=1}^{t}\prod_{j=i+1}^{t}(1 - \tilde{\alpha}_j(s, a))\tilde{\alpha}_i(s, a) \le 1.$$

*Proof.* Based on the definition of $\tau_i(s, a)$, we have

$$\sum_{i=1}^{t}\prod_{j=i+1}^{t}(1 - \tilde{\alpha}_j(s, a))\tilde{\alpha}_i(s, a) = \sum_{i \in T_1^t(s,a)}\prod_{j \in T_{i+1}^t(s,a)}(1 - \alpha\beta_j)\alpha\beta_i$$

$$= \sum_{i=1}^{|T_1^t(s,a)|}\prod_{j \in T_{t_i+1}^t(s,a)}(1 - \alpha\beta_j)\alpha\beta_{t_i},$$

where $t_i$ denotes the time stamp when $(s, a)$ is visited for the $i^{th}$ time in the window $[1, t]$.

Suppose there are $m \in [0, |T_1^t(s, a)|]$ non-zero $\beta_i$'s in the set $\left\{\beta_{t_1}, \ldots, \beta_{t_{|T_1^t(s,a)|}}\right\}$, i.e., $\sum_{i=1}^{|T_1^t(s,a)|}\beta_{t_i} = m$. Then we have

$$\sum_{i=1}^{t}\prod_{j=i+1}^{t}(1 - \tilde{\alpha}_j(s, a))\tilde{\alpha}_i(s, a) = \sum_{i=1}^{|T_1^t(s,a)|}\prod_{j \in T_{t_i+1}^t(s,a)}(1 - \alpha\beta_j)\alpha\beta_{t_i}$$

$$= \sum_{i=1}^{m}(1 - \alpha)^{m-i}\alpha$$

$$\le 1.$$

Since the above bound holds for any $|T_1^t(s, a)|$ and any $m \in [0, |T_1^t(s, a)|]$, we conclude the proof. □

**Proof of Theorem 2:**

Differently from the proof of Theorem 1 that does per iteration analysis, the central idea to analyze the asynchronous case is to capture the learning error in terms of the key noise and error terms over all the preceding iterations. Another novel development lies in the new method for analyzing the noise and error terms that involve the Bernoulli switching parameters specifically in double Q-learning.

Such a new analysis approach plays a critical role in improving the complexity bound in terms of its dependence on the sampling related parameters such as $L$ in (15) or $t_{\mathrm{mix}}$ in (16).

For the ease of presentation, we define the following notation, which denotes the index set of the iterations at which the state-action pair $(s, a)$ is updated.

**Definition 1.** *We denote by $T(s, a)$ the set of all the iteration indices at which the state-action pair $(s, a)$ is updated for either Q-estimator $Q^A$ or $Q^B$. In addition, we denote by $T_{t_1}^{t_2}(s, a) \subseteq T(s, a)$ the set of indices that are between time $t_1$ and $t_2$, that is,*

$$T_{t_1}^{t_2}(s, a) = \{t : t \in [t_1, t_2] \text{ and } t \in T(s, a)\}.$$

*The number of iterations updating $(s, a)$ between time $t_1$ and $t_2$ is thus given by $|T_{t_1}^{t_2}(s, a)|$, i.e., the cardinally of $T_{t_1}^{t_2}(s, a)$.*

Based on Definition 1, it is easy to observe that $\tau_t(s, a)$ in (3) can be rewritten as

$$\tau_t(s, a) = \mathbb{1}_{t \in T(s,a)}.$$

In addition, we keep the notations $r_t = Q_t^A - Q^*, \nu_t = Q_t^B - Q_t^A$.

Our proof proceeds with five steps as follows.

**Step I: Deriving a template bound.**

We first continue with the dynamics of $r_t(s, a)$ derived in Appendix A to characterize the error over all the preceding iterations, and obtain

$$
\begin{aligned}
r_{t+1}(s, a) &= (1 - \tilde{\alpha}_t(s, a))r_t(s, a) + \tilde{\alpha}_t(s, a)\left(\widehat{\mathcal{T}}_t Q_t^A(s, a) - \widehat{\mathcal{T}}_t Q^*(s, a)\right) \\
&\quad + \tilde{\alpha}_t(s, a)\varepsilon_t(s, a) + \tilde{\alpha}_t(s, a)\gamma\nu_t(s', a^*) \\
&= \prod_{i=1}^{t}(1 - \tilde{\alpha}_i(s, a))r_1(s, a) + \sum_{i=1}^{t}\prod_{j=i+1}^{t}(1 - \tilde{\alpha}_j(s, a))\tilde{\alpha}_i(s, a)\varepsilon_i(s, a) \\
&\quad + \sum_{i=1}^{t}\prod_{j=i+1}^{t}(1 - \tilde{\alpha}_j(s, a))\tilde{\alpha}_i(s, a)\left(\widehat{\mathcal{T}}_t Q_t^A(s, a) - \widehat{\mathcal{T}}_t Q^*(s, a)\right) \\
&\quad + \sum_{i=1}^{t}\prod_{j=i+1}^{t}(1 - \tilde{\alpha}_j(s, a))\tilde{\alpha}_i(s, a)\gamma\nu_i(s', a^*).
\end{aligned}
$$

Then we have

$$
\begin{aligned}
|r_{t+1}(s, a)| &= \left|\prod_{i=1}^{t}(1 - \tilde{\alpha}_i(s, a))r_1(s, a) + \sum_{i=1}^{t}\prod_{j=i+1}^{t}(1 - \tilde{\alpha}_j(s, a))\tilde{\alpha}_i(s, a)\varepsilon_i(s, a)\right. \\
&\quad + \sum_{i=1}^{t}\prod_{j=i+1}^{t}(1 - \tilde{\alpha}_j(s, a))\tilde{\alpha}_i(s, a)\left(\widehat{\mathcal{T}}_t Q_t^A(s, a) - \widehat{\mathcal{T}}_t Q^*(s, a)\right) \\
&\quad + \left.\sum_{i=1}^{t}\prod_{j=i+1}^{t}(1 - \tilde{\alpha}_j(s, a))\tilde{\alpha}_i(s, a)\gamma\nu_i(s', a^*)\right| \\
&\leq \underbrace{\prod_{i=1}^{t}(1 - \tilde{\alpha}_i(s, a))\|r_1\|}_{P_{1,t}(s,a)} + \underbrace{\left|\sum_{i=1}^{t}\prod_{j=i+1}^{t}(1 - \tilde{\alpha}_j(s, a))\tilde{\alpha}_i(s, a)\varepsilon_i(s, a)\right|}_{P_{2,t}(s,a)} \\
&\quad + \sum_{i=1}^{t}\prod_{j=i+1}^{t}(1 - \tilde{\alpha}_j(s, a))\tilde{\alpha}_i(s, a)\gamma\|r_i\| \\
&\quad + \sum_{i=1}^{t}\prod_{j=i+1}^{t}(1 - \tilde{\alpha}_j(s, a))\tilde{\alpha}_i(s, a)\gamma\|\nu_i\|. \quad (36)
\end{aligned}
$$

The next three steps will analyze the first two terms as well as $\|\nu_i\|$ in eq. (36), respectively.

**Step II: Bounding $\|P_{1,t}\|$.**

In this step, we prove a high probability bound for the term $P_{1,t}(s,a)$ in eq. (36). Note that $\tilde{\alpha}_t(s,a) = \alpha\beta_t\tau_t(s,a)$ is either 0 or $\alpha$. Thus the key to bound $\|P_{1,t}\|$ is to capture how many times $(s,a)$ is sampled to update $Q^A$ between $[1,t]$. To this end, we construct an auxiliary Markov chain with augmented states $\{s_t, a_t, \beta_t\}$, and use it to derive a concentration inequality for the sequence $\beta_t\tau_t(s,a)$ (see Lemma 5). Then the following high probability bound readily follows from Lemma 5.

**Proposition 4.** *Fix any $\delta \in (0,1)$ and $T > 0$ satsifying $T > \frac{886t_{\text{mix}}}{\mu_{\min}}\ln\left(\frac{4DT}{\delta}\right) := t_{\text{frame}}$ where $D = |\mathcal{S}||\mathcal{A}|$. Suppose Assumption 1 holds. Then with probability at least $1 - \delta$, we have*

$$\|P_{1,t}\| \leq (1-\alpha)^{\frac{1}{2}t\mu_{\min}}\|r_1\|, \tag{37}$$

*holds simultaneously for all $t$ satisfying $t_{\text{frame}} \leq t \leq T$.*

**Step III: Bounding $\|P_{2,t}\|$.**

In this step, we carefully analyze the coefficient of $P_{2,t}$ consisting of the learning rates, which is the key to keep the dependence order on the sampling related parameters tight. The following proposition provides the bound for term $P_{2,t}(s,a)$.

**Proposition 5.** *Fix any $\delta \in (0,1)$. Then with probability at least $1 - \delta$, we have*

$$\|P_{2,t}\| \leq \sqrt{2\alpha\ln\left(\frac{2DT}{\delta}\right)}V_{\max}, \tag{38}$$

*holds simultaneously for all $t \in [1,T]$, where $D = |\mathcal{S}||\mathcal{A}|$.*

*Proof.* See Appendix G. $\qquad\square$

**Step IV: Bounding $\|\nu_t\|$.**

The following proposition provides the bound on $\|\nu_t\|$.

**Proposition 6.** *Fix any $\delta \in (0,1)$. Then with probability at least $1 - \delta$, we have*

$$\|\nu_t\| \leq 3\sqrt{2\alpha\ln\left(\frac{2DT}{\delta}\right)}\frac{V_{\max}}{1-\gamma}, \tag{39}$$

*holds simultaneously for all $t \in [1,T]$, where $D = |\mathcal{S}||\mathcal{A}|$.*

*Proof.* See Appendix H. $\qquad\square$

**Step V: Overall convergence.**

In this final step, we apply the above propositions to (36) and obtain that, with probability at least $1 - 3\delta$, the following holds simultaneously for all $(s,a)$-pair and all $t$ satisfying $t_{\text{frame}} \leq t \leq T$,

$$
\begin{aligned}
|r_{t+1}(s,a)| &\leq \prod_{i=1}^{t}(1-\tilde{\alpha}_i(s,a))\|r_1\| + \left|\sum_{i=1}^{t}\prod_{j=i+1}^{t}(1-\tilde{\alpha}_j(s,a))\tilde{\alpha}_i(s,a)\varepsilon_i(s,a)\right| \\
&\quad + \sum_{i=1}^{t}\prod_{j=i+1}^{t}(1-\tilde{\alpha}_j(s,a))\tilde{\alpha}_i(s,a)\gamma\|r_i\| \\
&\quad + \sum_{i=1}^{t}\prod_{j=i+1}^{t}(1-\tilde{\alpha}_j(s,a))\tilde{\alpha}_i(s,a)\gamma\|\nu_i\| \\
&\leq (1-\alpha)^{\frac{1}{2}t\mu_{\min}}\|r_1\| + \sum_{i=1}^{t}\prod_{j=i+1}^{t}(1-\tilde{\alpha}_j(s,a))\tilde{\alpha}_i(s,a)\gamma\|r_i\|
\end{aligned}
$$

$$+ \sqrt{2\alpha \ln \left( \frac{2DT}{\delta} \right)} V_{\max} + 3\sqrt{2\alpha \ln \left( \frac{2DT}{\delta} \right)} \frac{\gamma V_{\max}}{1-\gamma} \sum_{i=1}^{t} \prod_{j=i+1}^{t} (1 - \tilde{\alpha}_j(s,a)) \tilde{\alpha}_i(s,a)$$

$$\overset{(i)}{\leq} (1-\alpha)^{\frac{1}{2} t \mu_{\min}} \|r_1\| + \sum_{i=1}^{t} \prod_{j=i+1}^{t} (1 - \tilde{\alpha}_j(s,a)) \tilde{\alpha}_i(s,a) \gamma \|r_i\|$$

$$+ \sqrt{2\alpha \ln \left( \frac{2DT}{\delta} \right)} V_{\max} + 3\sqrt{2\alpha \ln \left( \frac{2DT}{\delta} \right)} \frac{\gamma V_{\max}}{1-\gamma}$$

$$\triangleq (1-\alpha)^{\frac{1}{2} t \mu_{\min}} \|r_1\| + \sum_{i=1}^{t} \prod_{j=i+1}^{t} (1 - \tilde{\alpha}_j(s,a)) \tilde{\alpha}_i(s,a) \gamma \|r_i\| + C, \tag{40}$$

where (i) follows from Lemma 6, and the last equality follows from the following definition,

$$C := \sqrt{2\alpha \ln \left( \frac{2DT}{\delta} \right)} V_{\max} + 3\sqrt{2\alpha \ln \left( \frac{2DT}{\delta} \right)} \frac{\gamma V_{\max}}{1-\gamma}. \tag{41}$$

We further define the following quantities for the ease of presentation:

$$\mu_{\text{frame}} := \frac{1}{2} \mu_{\min} t_{\text{frame}}, \tag{42}$$

$$\rho := (1-\gamma) \left( 1 - (1-\alpha)^{\mu_{\text{frame}}} \right), \tag{43}$$

where $t_{\text{frame}}$ is defined in Proposition 4.

The following proposition follows from a direct application of Li et al. (2020, Lemmas 3 and 4) to the $r_t$ bound of the double Q-learning in (40). Note that the constants $C, t_{\text{frame}}$ here are of different values from those in Li et al. (2020, Lemmas 3 and 4).

**Proposition 7.** *Suppose the inequality dynamics of $r_t$ in (40) holds. For any $\delta \in (0, \frac{1}{4}), \epsilon \in (0, \frac{1}{(1-\gamma)}]$, then with probability at least $1 - 4\delta$, we have*

$$\|r_t\| \leq \frac{C}{1-\gamma} + (1-\rho)^k \frac{\|r_1\|}{1-\gamma} + \epsilon, \tag{44}$$

*where $C$ is defined in (41), $k = \max \left\{ 0, \lfloor \frac{t - t_{\text{th}}}{t_{\text{frame}}} \rfloor \right\}$, and $t_{\text{th}} := \max \left\{ \frac{2 \ln \frac{1}{(1-\gamma)\epsilon}}{\alpha \mu_{\min}}, t_{\text{frame}} \right\}$.*

Now to derive the time complexity, first, it is easy to verify that $\frac{C}{1-\gamma} \leq \epsilon$ by choosing

$$\alpha \leq \frac{(1-\gamma)^6 \epsilon^2}{32 \ln \frac{2DT}{\delta}}. \tag{45}$$

Next, we have $(1-\rho)^k \frac{\|r_1\|}{1-\gamma} \leq \exp(-\rho k) \frac{\|r_1\|}{1-\gamma} \leq \epsilon$ if $k \geq \ln \frac{\|r_1\|}{(1-\gamma)\epsilon} / \rho$. Recalling the definition of $k$ in Proposition 7, we solve for $t$ and obtain

$$t \geq t_{\text{th}} + t_{\text{frame}} + \frac{t_{\text{frame}}}{\rho} \ln \frac{\|r_1\|}{(1-\gamma)\epsilon}. \tag{46}$$

Further, by the Bernoulli's inequality, we have $(1-\alpha)^{\mu_{\text{frame}}} \leq 1 - \frac{\alpha \mu_{\text{frame}}}{2}$ if $\alpha < \frac{1}{\mu_{\text{frame}} - 1}$. Then we have

$$\rho = (1-\gamma) \left( 1 - (1-\alpha)^{\mu_{\text{frame}}} \right) \geq \frac{\alpha \mu_{\text{frame}} (1-\gamma)}{2}. \tag{47}$$

Last, we derive an upper bound on the RHS of (46),

$$\text{RHS of (46)} \overset{(i)}{\leq} t_{\text{th}} + t_{\text{frame}} + \frac{2t_{\text{frame}}}{\alpha\mu_{\text{frame}}(1-\gamma)}\ln\frac{\|r_1\|}{(1-\gamma)\epsilon}$$

$$\overset{(ii)}{=} t_{\text{th}} + t_{\text{frame}} + \frac{4}{\alpha\mu_{\min}(1-\gamma)}\ln\frac{\|r_1\|}{(1-\gamma)\epsilon}$$

$$\overset{(iii)}{\leq} t_{\text{th}} + t_{\text{frame}} + \frac{4}{\mu_{\min}(1-\gamma)}\ln\frac{\|r_1\|}{(1-\gamma)\epsilon}\cdot\max\left\{\frac{32\ln\frac{2DT}{\delta}}{(1-\gamma)^6\epsilon^2}, \mu_{\text{frame}}\right\}$$

$$= t_{\text{th}} + t_{\text{frame}} + \frac{4}{\mu_{\min}(1-\gamma)}\ln\frac{\|r_1\|}{(1-\gamma)\epsilon}\cdot\max\left\{\frac{32\ln\frac{2DT}{\delta}}{(1-\gamma)^6\epsilon^2}, 443t_{\text{mix}}\ln\frac{4DT}{\delta}\right\}.$$

where (i) follows from (47), (ii) follows from the definition (42), and (iii) follows from the bounds (45) and $\alpha \leq \frac{1}{\mu_{\text{frame}}}$. Thus, continuing with (44), we conclude that for any $\delta \in (0, 1/7)$, with probability at least $1 - 7\delta$, we have $\|r_T\| \leq 3\epsilon$ as long as

$$T = \tilde{\Omega}\left(\frac{1}{\mu_{\min}\epsilon^2(1-\gamma)^7}\ln\frac{1}{\epsilon(1-\gamma)^2} + \frac{t_{\text{mix}}}{\mu_{\min}(1-\gamma)}\ln\frac{1}{\epsilon(1-\gamma)^2}\right).$$

$\square$

# G   Proof of Proposition 5

We first provide a useful lemma, which provides a bound on the summation of a sequence of discounted random variables (not necessarily independent).

**Lemma 7.** *Fix $k > 0$ and $\alpha \in (0, 1)$. Given a sequence of random variables $\{X_i\}$ and a filtration $\{\mathcal{F}_i\}$ satisfying $\mathbb{E}(X_i|\mathcal{F}_i) = 0$ and $|X_i| \leq \bar{c}$, then for any $w > 0$,*

$$\mathbb{P}\left(\left|\sum_{i=1}^k (1-\alpha)^{k-i}\alpha X_i\right| \geq w\right) \leq 2\exp\left(-\frac{w^2}{2\alpha\bar{c}^2}\right).$$

*Proof.* Define $\{M_i\}_{1\leq i \leq k}$ as

$$M_{i+1} = (1-\alpha)M_i + \alpha X_i, \qquad \text{with } M_1 = 0.$$

Clearly we have $M_{k+1} = \sum_{i=1}^k (1-\alpha)^{k-i}\alpha X_i$, and

$$\begin{aligned}
\mathbb{E}(M_{i+1}|\mathcal{F}_i) &= \mathbb{E}((1-\alpha)M_i + \alpha X_i|\mathcal{F}_i)\\
&= (1-\alpha)M_i + \mathbb{E}(\alpha X_i|\mathcal{F}_i)\\
&= (1-\alpha)M_i.
\end{aligned}$$

Next, we construct $\{\tilde{M}_i\}$ as

$$\tilde{M}_i := (1-\alpha)^{k-i+1}M_i, \quad 1 \leq i \leq k+1.$$

Then $\{\tilde{M}_i\}_{1\leq i \leq k+1}$ is a martingale sequence with $\tilde{M}_{k+1} = M_{k+1}$ and $\mathbb{E}\left(\tilde{M}_1\right) = 0$. We refer to $\{\tilde{M}_i\}_{1\leq i \leq k+1}$ as the *martingale surrogate* of $\{M_i\}_{1\leq i \leq k+1}$.

Further observe that

$$d_i := \left|\tilde{M}_{i+1} - \tilde{M}_i\right| \leq (1-\alpha)^{k-i}M_{i+1} - (1-\alpha)^{k-i+1}M_i = (1-\alpha)^{k-i}\alpha X_i, \quad 1 \leq i \leq k.$$

Then it follows that

$$d_i^2 \leq (1-\alpha)^{2(k-i)}\alpha^2|X_i|^2 \leq (1-\alpha)^{k-i}\alpha^2\bar{c}^2,$$

where the last inequality follows because $(1-\alpha)^2 < 1-\alpha$ and $|X_i| \leq \bar{c}$.

Applying the Azuma-Hoeffding Inequality (see Lemma 3) yields

$$\mathbb{P}\left(\left|\sum_{i=1}^{k}(1-\alpha)^{k-i}\alpha X_i\right| \geq w\right) = \mathbb{P}\left(|M_{k+1}| \geq w\right)$$

$$\leq 2\exp\left(-\frac{w^2}{2\sum_{i=1}^{k}d_i^2}\right)$$

$$\leq 2\exp\left(-\frac{w^2}{2\sum_{i=1}^{k}(1-\alpha)^{k-i}\alpha^2\bar{c}^2}\right)$$

$$\leq 2\exp\left(-\frac{w^2}{2\alpha\bar{c}^2}\right).$$

$\square$

We now proceed the proof of Proposition 5 as follows. Recall that

$$P_{2,t}(s,a) = \left|\sum_{i=1}^{t}\prod_{j=i+1}^{t}(1-\tilde{\alpha}_j(s,a))\tilde{\alpha}_i(s,a)\varepsilon_i(s,a)\right|$$

$$= \left|\sum_{i\in T_1^t(s,a)}\prod_{j\in T_{i+1}^t(s,a)}(1-\alpha\beta_j)\alpha\beta_i\varepsilon_i(s,a)\right|$$

$$= \left|\sum_{i=1}^{|T_1^t(s,a)|}\prod_{j\in T_{t_i+1}^t(s,a)}(1-\alpha\beta_j)\alpha\beta_{t_i}\varepsilon_{t_i}(s,a)\right|,$$

where $t_i$ denotes the time stamp at which $(s,a)$ is sampled for the $i^{th}$ time in the window $[1,t]$.

It suffices to show that for any fixed $m := |T_1^t(s,a)| \in [0,t]$ and $w \in (0,1)$, we have

$$\mathbb{P}\left(P_{2,t}(s,a) \geq w\right) = \mathbb{P}\left(\left|\sum_{i=1}^{m}\prod_{j\in T_{t_i+1}^t(s,a)}(1-\alpha\beta_j)\alpha\beta_{t_i}\varepsilon_{t_i}(s,a)\right| \geq w\right)$$

$$\leq 2\exp\left(-\frac{w^2}{2\alpha V_{\max}^2}\right).$$

Then letting the upper bound to be $\frac{\delta}{DT}$ where $\delta \in (0,1)$, solving $w$, and further using the union bound will yield the desired result stated in Proposition 5.

To this end, we observe that

$$\mathbb{P}\left(P_{2,t}(s,a) \geq w\right) = \mathbb{P}\left(P_{2,t}(s,a) \geq w\Big|\sum_{i=1}^{m}\beta_{t_i} = 0\right)\mathbb{P}\left(\sum_{i=1}^{m}\beta_{t_i} = 0\right)$$

$$+ \mathbb{P}\left(P_{2,t}(s,a) \geq w\Big|\sum_{i=1}^{m}\beta_{t_i} = 1\right)\mathbb{P}\left(\sum_{i=1}^{m}\beta_{t_i} = 1\right)$$

$$+ \cdots$$

$$+ \mathbb{P}\left(P_{2,t}(s,a) \geq w\Big|\sum_{i=1}^{m}\beta_{t_i} = m\right)\mathbb{P}\left(\sum_{i=1}^{m}\beta_{t_i} = m\right). \qquad (48)$$

For any $k \in [0, m]$, we have

$$\mathbb{P}\left(P_{2,t}(s,a) \geq w \Big| \sum_{i=1}^{m} \beta_{t_i} = k\right)$$

$$= \mathbb{P}\left(\left|\sum_{i=1}^{m} \prod_{j \in T_{t_i+1}^{j}(s,a)} (1 - \alpha\beta_j)\alpha\beta_{t_i}\varepsilon_{t_i}(s,a)\right| \geq w \Big| \sum_{i=1}^{m} \beta_{t_i} = k\right)$$

$$\stackrel{\text{(i)}}{=} \mathbb{P}\left(\left|\sum_{i=1}^{k}(1-\alpha)^{k-i}\alpha\varepsilon_{t_i'}(s,a)\right| \geq w\right)$$

$$\stackrel{\text{(ii)}}{\leq} 2\exp\left(-\frac{w^2}{2\alpha V_{\max}^2}\right), \tag{49}$$

where in (i) $t_i'$ denotes the time stamp of the $i^{th}$ non-zero $\beta_{t_i}$ in the sequential array $(\beta_{t_1}, \beta_{t_2}, \ldots, \beta_{t_m})$, and (ii) follows from Lemma 7 with the fact that $\mathbb{E}(\varepsilon_i(s,a)) = 0$ and $|\varepsilon_i(s,a)| \leq V_{\max}$.

Thus, substituting eq. (49) into eq. (48), we obtain

$$\mathbb{P}(P_{2,t}(s,a) \geq w) \leq 2\exp\left(-\frac{w^2}{2\alpha V_{\max}^2}\right)\sum_{k=0}^{m}\mathbb{P}\left(\sum_{i=1}^{m}\beta_{t_i} = k\right) = 2\exp\left(-\frac{w^2}{2\alpha V_{\max}^2}\right), \tag{50}$$

which completes the proof.

## H    Proof of Proposition 6

We first prove a useful lemma.

**Lemma 8.** *Fix $\delta \in (0,1)$. With probability at least $1 - \delta$, the following inequality holds simultaneously for all $(s,a)$ and all $t \in [1, T]$,*

$$\left|\sum_{i=1}^{t}\prod_{j=i+1}^{t}\left(1 - \frac{\hat{\alpha}_j(s,a)}{2}\right)\hat{\alpha}_i(s,a)\mu_i(s,a)\right| \leq 3\sqrt{2\alpha\ln\left(\frac{2DT}{\delta}\right)}V_{\max}, \tag{51}$$

*where $D = |\mathcal{S}||\mathcal{A}|$.*

*Proof.* Observe that

$$\sum_{i=1}^{t}\prod_{j=i+1}^{t}\left(1 - \frac{\hat{\alpha}_j(s,a)}{2}\right)\hat{\alpha}_i(s,a)\mu_i(s,a)$$

$$= \sum_{i \in T_1^t(s,a)}\left(1 - \frac{\alpha}{2}\right)^{|T_1^t(s,a)|-i}\alpha\mu_i(s,a)$$

$$= \sum_{i=1}^{|T_1^t(s,a)|}\left(1 - \frac{\alpha}{2}\right)^{|T_1^t(s,a)|-i}\alpha\mu_{t_i}(s,a),$$

where $t_i$ denotes the time stamp when $(s,a)$ is sampled for the $i^{th}$ time in the window $[1, t]$.

It suffices to show that for any $m = |T_1^t(s,a)| \in [0, t]$, we have

$$\mathbb{P}\left(\left|\sum_{i=1}^{t}\prod_{j=i+1}^{t}\left(1 - \frac{\hat{\alpha}_j(s,a)}{2}\right)\hat{\alpha}_i(s,a)\mu_i(s,a)\right| \geq w\right)$$

$$= \mathbb{P}\left(\left|\sum_{i=1}^{m}\left(1 - \frac{\alpha}{2}\right)^{m-i}\alpha\mu_{t_i}(s,a)\right| \geq w\right)$$

$$\leq 2\exp\left(-\frac{w^2}{18\alpha V_{\max}^2}\right) := \frac{\delta}{DT}.$$

where the last inequality follows from Lemma 7 by observing that $|\mu_i(s,a)| \leq 3V_{\max}$ and $\mathbb{E}(\mu_{t_i}(s,a)|\mathcal{F}'_i) = \mathbb{E}(\mu_{t_i}(s,a)|\mathcal{F}_{t_i}) = 0$ as derived in Appendix A. Thus, letting the probability be bounded by $\frac{\delta}{DT}$ and using the union bound over all $(s,a)$-pair and all $t \in [1,T]$, we complete the proof.

$\square$

We now proceed the proof of Proposition 6 by starting with the dynamics of $\nu_t$ derived in Appendix A, and have

$$
\begin{aligned}
\nu_{t+1}(s,a) &= Q^B_{t+1}(s,a) - Q^A_{t+1}(s,a) \\
&= (1 - \hat{\alpha}_t(s,a))\nu_t(s,a) + \hat{\alpha}_t(s,a)\mathcal{H}_t(s,a) + \hat{\alpha}_t(s,a)\mu_t(s,a) \\
&= (1 - \hat{\alpha}_t(s,a))\nu_t(s,a) + \hat{\alpha}_t(s,a)\left( \frac{1}{2}\nu_t(s,a) + \frac{\gamma}{2}\underbrace{\mathbb{E}_{s'}\left(Q^B_t(s',a^*) - Q^A_t(s',b^*)\right)}_{J_t(s,a)} \right) \\
&\quad + \hat{\alpha}_t(s,a)\mu_t(s,a) \\
&= \left(1 - \frac{\hat{\alpha}_t(s,a)}{2}\right)\nu_t(s,a) + \frac{\gamma\hat{\alpha}_t(s,a)}{2}J_t(s,a) + \hat{\alpha}_t(s,a)\mu_t(s,a) \\
&= \prod_{i=1}^t \left(1 - \frac{\hat{\alpha}_i(s,a)}{2}\right)\nu_1(s,a) + \sum_{i=1}^t \prod_{j=i+1}^t \left(1 - \frac{\hat{\alpha}_j(s,a)}{2}\right)\frac{\gamma\hat{\alpha}_i(s,a)}{2}J_i(s,a) \\
&\quad + \sum_{i=1}^t \prod_{j=i+1}^t \left(1 - \frac{\hat{\alpha}_j(s,a)}{2}\right)\hat{\alpha}_i(s,a)\mu_i(s,a) \\
&\overset{\text{(i)}}{=} \sum_{i=1}^t \prod_{j=i+1}^t \left(1 - \frac{\hat{\alpha}_j(s,a)}{2}\right)\frac{\gamma\hat{\alpha}_i(s,a)}{2}J_i(s,a) \\
&\quad + \sum_{i=1}^t \prod_{j=i+1}^t \left(1 - \frac{\hat{\alpha}_j(s,a)}{2}\right)\hat{\alpha}_i(s,a)\mu_i(s,a),
\end{aligned}
$$

where (i) follows because $\|\nu_1\| = 0$.

Next, we have

$$
\begin{aligned}
|\nu_{t+1}(s,a)| &\leq \left| \sum_{i=1}^t \prod_{j=i+1}^t \left(1 - \frac{\hat{\alpha}_j(s,a)}{2}\right)\frac{\hat{\alpha}_i(s,a)}{2}\gamma J_i(s,a) \right| \\
&\quad + \left| \sum_{i=1}^t \prod_{j=i+1}^t \left(1 - \frac{\hat{\alpha}_j(s,a)}{2}\right)\hat{\alpha}_i(s,a)\mu_i(s,a) \right| \\
&\overset{\text{(i)}}{\leq} \sum_{i=1}^t \prod_{j=i+1}^t \left(1 - \frac{\hat{\alpha}_j(s,a)}{2}\right)\frac{\hat{\alpha}_i(s,a)}{2}\gamma \|\nu_i\| \\
&\quad + \left| \sum_{i=1}^t \prod_{j=i+1}^t \left(1 - \frac{\hat{\alpha}_j(s,a)}{2}\right)\hat{\alpha}_i(s,a)\mu_i(s,a) \right|, \tag{52}
\end{aligned}
$$

where (i) follows from the property $|J_i(s,a)| \leq \|\nu_i\|$ derived in Appendix A.

Next, following from Lemma 8, we have with probability at least $1 - \delta$ that the following inequality holds simultaneously for all $t \in [1,T]$ and all $(s,a)$-pair,

$$
|\nu_{t+1}(s,a)| \leq \sum_{i=1}^t \prod_{j=i+1}^t \left(1 - \frac{\hat{\alpha}_j(s,a)}{2}\right)\frac{\hat{\alpha}_i(s,a)}{2}\gamma \|\nu_i\| + 3\sqrt{2\alpha \ln\left(\frac{2DT}{\delta}\right)}V_{\max}.
$$

We finally complete the proof by induction. For the ease of presentation, denote $b :=$ $3\sqrt{2\alpha \ln\left(\frac{2DT}{\delta}\right)} \frac{V_{\max}}{1-\gamma}$. The base case holds trivially since $\|\nu_1\| = 0$. Suppose that $\|\nu_t\| \leq b$ for any $t \geq 2$. Then for the case of $t + 1$, we have

$$\|\nu_{t+1}\| \leq \sum_{i=1}^{t} \prod_{j=i+1}^{t} \left(1 - \frac{\hat{\alpha}_j(s,a)}{2}\right) \frac{\hat{\alpha}_i(s,a)}{2} \gamma \|\nu_i\| + (1-\gamma)b$$

$$\leq \sum_{i=1}^{t} \prod_{j=i+1}^{t} \left(1 - \frac{\hat{\alpha}_j(s,a)}{2}\right) \frac{\hat{\alpha}_i(s,a)}{2} \gamma b + (1-\gamma)b$$

$$\overset{(i)}{\leq} \gamma b + (1-\gamma)b$$

$$= b,$$

where (i) follows from Lemma 6 by replacing $\beta_i = \frac{1}{2}$ which does not affect the upper bound.