# OpenReview forum: "Faster Non-asymptotic Convergence for Double Q-learning"
_NeurIPS.cc/2021/Conference — NeurIPS 2021 Poster_

### Official Review · Reviewer_fiWX · 2021-07-11

**Rating:** 6
**Confidence:** 2

**Summary:**

This paper establishes tighter finite-time bounds for double Q-learning with constant learning rates in both synchronous
and asynchronous Q-learning settings.
A major difference between this paper and related work is that it only requires a constant learning rate to achieve fast convergence.

**Limitations And Societal Impact:**

I am not an expert in reinforcement learning theory. My questions focus on how the given results can impact on practice. I am willing to increase my score if my concerns are addressed.

(1) **(My major concern)** The main purpose of using double Q-learning is to reduce overestimation bias in value estimation. Does the analysis reveal any new insights on the reduction of overestimation? e.g., comparing to the analysis of vanilla Q-learning, I am curious about whether their error propagation behaviors are different. Does the reduction of overestimation can explicitly or implicitly provide any good properties for learning stability.

(2) Line 341mentions that the current convergence analysis of double Q-learning has not achieved the same rate as that of vanilla Q-learning. How does their convergence performance compare in empirical evaluations with different learning rate scheduling mechanisms? I am curious about the literature, some discussions about related results are helpful. In addition, you can extend Table 1 to include prior results on vanilla Q-learning with different learning rate scheduling (maybe in Appendix).

(3) In the asynchronous setting, whether the time complexity can translate to sample complexity? If so, why the proposed analysis does not rely on an exploration mechanism? To my knowledge, the fast convergence of tabular Q-learning relies on specific exploration strategies such as UCB exploration. Are there any assumptions on the chosen state-action pairs for updating?

**Main Review:**

**Originality**: I am not an expert in the theory area. To my knowledge, the proposed problem formulation and analysis are different from the related work discussed in this paper, which becomes more realistic.

**Quality and Clarity**: The structure of this paper is well-organized. The discussion of preliminaries and related work is suitable.

**Significance**: This paper aims to establish a better theoretical characterization of double Q-learning, which is a widely used technique in empirical research. I agree this paper studies a very important problem in reinforcement learning. To my knowledge, removing algorithmic assumptions on learning scheduling should have solid contributions to the related literature, since the constant learning rate setting is more realistic than related work and may serve some extent of insights for function approximation settings. I vote for *weak accept* since I have some concerns about the connections to practical scenarios (see questions).

**Time Spent Reviewing:**

5 hours

---

> ### Author Response · Authors · 2021-08-10
> **Many thanks for the expert review!**
>
> **Q(1): (Major concern)** The main purpose of using double Q-learning is to reduce overestimation bias in value estimation. Does the analysis reveal any new insights on the reduction of overestimation? e.g., comparing to the analysis of vanilla Q-learning, I am curious about whether their error propagation behaviors are different. Does the reduction of overestimation can explicitly or implicitly provide any good properties for learning stability.
>
> **A:** Thanks for all these good questions! Our analysis of the two nested SAs (stochastic approximations) provides further insights on the reduction of overestimation: high-level speaking, our finite-time characterizations show that the convergence rate of $\Vert Q^{A}-Q^{\*} \Vert $ is limited by that of $\Vert Q^{A}-Q^{B} \Vert $. This suggests that neither $Q^{A}$ nor $Q^{B}$ can approach $Q^{\*}$ alone too aggressively, which implies the mitigation of overestimation. This is also the major difference from the error propagation of Q-learning.
> Intuitively, the reduction of overestimation will reduce the stochastic bias of the algorithm, which will yield more stable convergence. Experimentally, this has been demonstrated in (Hasselt et al., 2016) through extensive simulation studies in comparison with Q-learning.
>
> **Q(2):** Line 341 mentions that the current convergence analysis of double Q-learning has not achieved the same rate as that of vanilla Q-learning. How does their convergence performance compare in empirical evaluations with different learning rate scheduling mechanisms? I am curious about the literature, some discussions about related results are helpful. In addition, you can extend Table 1 to include prior results on vanilla Q-learning with different learning rate scheduling (maybe in Appendix).
>
> **A:** Thanks for the suggestions. Experimentally, in the literature, there is no conclusive comparison between Q-learning and Double Q-learning. For example, a recent experimental study (Weng, et al, 2020) under a similar setting to this paper shows that when using the same step size, Q-learning has a faster rate of convergence initially but suffers from a higher mean-squared error. This observation matches our theoretical analysis. On the other hand,  when double Q-learning uses twice the step size of Q-learning, their simulation shows that double Q-learning achieves a faster initial convergence rate, at the cost of a possibly worse mean-squared error than Q-learning. This topic is still up to the future investigation to have a solid conclusion.
>
> We will add more discussions of the above points in our revision. We will also include the convergence rate comparison with vanilla Q-learning in the Appendix, as suggested.
>
>
> **Q(3):** In the asynchronous setting, whether the time complexity can translate to sample complexity? If so, why the proposed analysis does not rely on an exploration mechanism? To my knowledge, the fast convergence of tabular Q-learning relies on specific exploration strategies such as UCB exploration. Are there any assumptions on the chosen state-action pairs for updating?
>
> **A:** Great question! First, in our asynchronous setting, the time complexity is equivalent to sample complexity, because each iteration samples one state-action pair.
>
> Our analysis does not rely on any specific exploration mechanism, such as  $\epsilon$-greedy or UCB exploration. Instead, the exploration is implicitly considered by our assumption (see Assumption 1) about how likely each state-action pair is sampled beyond a certain range of time. This is a standard assumption commonly adopted by theoretical analysis of Q-learning algorithms. Under such an assumption, the exploration strategy is captured by the parameters $\mu_{\min}$ and $t_{\text{mix}}$, i.e., roughly speaking, each state-action pair has at least a positive probability $\mu_{\min}$ to be visited after about $t_{\text{mix}}$ steps/iterations. Namely, larger $\mu_{\min}$ and smaller $t_{\text{mix}}$ correspond to more exploration. Our Theorem 2 characterizes how the time/sample complexity depends on these two parameters, which indicates that larger $\mu_{\min}$ and smaller $t_{\text{mix}}$ (i.e., more exploration) will result in lower time/sample complexity.
>
> **References**
>
> (Hasselt et al., 2016) Hado van Hasselt , Arthur Guez, and David Silver. Deep Reinforcement Learning with Double Q-Learning. Proceedings of the Thirtieth AAAI Conference on Artificial Intelligence (AAAI-16)
>
> (Weng et al., 2020) Wentao Weng, Harsh Gupta, Niao He, Lei Ying, R. Srikant. The Mean-Squared Error of Double Q-Learning, NeurIPS, 2020

---

### Official Review · Reviewer_cHVh · 2021-07-15

**Rating:** 6
**Confidence:** 5

**Summary:**

The paper provides new error bounds for double Q-learning using constant stepsize, which improves over existing result based on polynomial stepsize.  My only concern is that there are limited insights from this paper about double Q-learning. See detailed comments below.

**Main Review:**

Strength:
- (Originality) The finite-time bounds under constant stepsize for double Q-learning are new and improve over existing paper by Xiong et al., 2020 which uses the polynomial stepsize.  The result follows by analyzing the error dynamics of Q_A-Q^* and Q_A-Q_B based on their stochastic approximation recursions.

- (Clarity) The  paper for the most part is clearly written and easy to follow. Comparisons to related work are carefully discussed.

- (Quality) Technical contribution is good although not particularly strong. The authors justified the difference in the analysis from Xiong et al, 2020,  but to a large extent, the analysis is more or less adapted from the existing Q-learning analysis in (Wainwright, 2019) and  (Li et al, 2020) for synchronous  ​and asynchronous settings.

Weakness:
- (Significance)  The main contribution of the paper is to show that using constant learning rate instead of polynomial rate yields better convergence of double Q-learning.  This of course is not too surprising given that similar results (e.g., Li et al, 2020) have already been established for Q-learning. I am not sure if this paper provides any more insights beyond this point. Given that the results only reflect the upper bounds (which may not necessarily  be tight), it certainly would be more convincing to provide some lower bound, or compare the exact behaviors, or at least validate the performance through numerical experiments.

- (Significance) A perhaps more significant research question is  how does the non-asymptotic bounds of double Q-learning compare to Q-learning when both are using the constant stepsize.  The bound in this paper indicates that double Q-learning has a worse complexity than Q-learning with a factor of O((1-\gamma)^{-2}) and requires a smaller stepsize than Q-learning.  But this deviates from what people normally observe in practice. It remains unclear whether the current bound is sharp or meaningful enough to shed light on the advantage of double Q-learning.

Minor Suggestions:
 In the proof sketch, Proposition 1 is applied several times, but it is only presented in the appending. The definitions of H_t and mu_t in the SA recursion (6) is never defined in the main text. The authors mentioned their technical novelty of using augmented Markov chain and conditional concentration analysis. It is hard to understand what are these and why they are needed without going through all the proofs in the appendix. Some extra efforts are needed to make the main text more self-contained.


**Time Spent Reviewing:**

10 hours

---

> ### Author Response · Authors · 2021-08-10
> **Many thanks for the expert review!**
>
> **Q1: (Significance)** The main contribution of the paper is to show that using constant learning rate instead of polynomial rate yields better convergence of double Q-learning. This of course is not too surprising given that similar results (e.g., Li et al, 2020) have already been established for Q-learning. I am not sure if this paper provides any more insights beyond this point. Given that the results only reflect the upper bounds (which may not necessarily be tight), it certainly would be more convincing to provide some lower bound, or compare the exact behaviors, or at least validate the performance through numerical experiments.
>
> **A:** Many thanks for the comment! Although the convergence rate for vanilla Q-learning under the constant learning rate has been characterized in (Li et. al., 2020), the analysis for double Q-learning is rather difficult due to the special challenges arising in double Q-learning. The key challenges are to deal with the switching random variable $\beta_t$ that captures the switching process between updating the two Q-estimators, and the inter-connection between two Q-estimators ($\nu_t$ in the paper), which are unique in the analysis of double Q-learning and need considerable effort. Thus, we find these analysis techniques developed in this paper to be new contributions to the community.
>
> In terms of lower bound, (Azar et al., 2013) has developed a lower bound on the sample complexity for finding optimal action-value function for tabular MDP, which is applicable here. Comparing with such a lower bound, our upper bound is tight in terms of its dependence on the major parameters: the learning error $\epsilon$ and the dimension $D$, whereas (Xiong et al., 2020) didn’t achieve such optimality. The only gap between our bound and the lower bound given in (Azar et al., 2013) is the dependence on $1-\gamma$, which is an interesting yet challenging topic for our future study.
>
> We have done a quick numerical experiment and showed that the experimental convergence rate of double Q-learning under the constant learning rate matches our theoretical convergence rate in terms of the dependence on the learning error $\epsilon$. We will include this experiment in the revision.
>
> **Q2: (Significance)** A perhaps more significant research question is how does the non-asymptotic bounds of double Q-learning compare to Q-learning when both are using the constant stepsize. The bound in this paper indicates that double Q-learning has a worse complexity than Q-learning with a factor of O((1-\gamma)^{-2}) and requires a smaller stepsize than Q-learning. But this deviates from what people normally observe in practice. It remains unclear whether the current bound is sharp or meaningful enough to shed light on the advantage of double Q-learning.
>
> **A:** Great point! To our understanding, the original design of double Q-learning is to reduce overestimation. Our analysis of the two nested SAs shed light on such an advantage of double Q-learning. High-level speaking, our finite-time characterizations show that the convergence rate of $\Vert Q^{A} - Q^{\*} \Vert $ is limited by that of $\Vert Q^{A}-Q^{B} \Vert $. This suggests that neither $Q^{A}$ nor $Q^{B}$ can approach to $Q^{\*}$ alone too aggressively, which implies the mitigation of overestimation.
>
> Experimentally, there is no conclusive comparison on the practical performance between double Q-learning and Q-learning. For example, a recent simulation study (Weng et al., 2020) under a similar setting to this paper shows that when using the same step size, Q-learning has a faster convergence rate than double Q-learning initially but suffers from a higher mean-squared error. This observation matches our theoretical analysis. On the other hand, when double Q-learning uses twice the step size of Q-learning, their simulation shows that double Q-learning achieves a faster initial convergence rate, at the cost of a possibly worse mean-squared error than Q-learning. This topic is still up to the future investigation to have a solid conclusion.
>
> Lastly, as commented in our response to Q1, our bound is tight in terms of the dependence on the major parameters, whereas the best existing results in (Xiong et al., 2020) do not have such optimality. Full understandings of the theoretical bound still need more investigation.
>
> **Q: (Minor Suggestions)** In the proof sketch, Proposition 1 is applied several times, but it is only presented in the appending. The definitions of $H_t$ and $\mu_t$ in the SA recursion (6) is never defined in the main text. The authors mentioned their technical novelty of using augmented Markov chain and conditional concentration analysis. It is hard to understand what are these and why they are needed without going through all the proofs in the appendix. Some extra efforts are needed to make the main text more self-contained.
>
> **A:** Many thanks for the suggestions. Due to the space limit, we were only able to keep the most important results in the main text. We will move the definitions of $H_t$ and $\mu_t$ to the main text and add more explanations of our proof to make it more self-contained.
>
> **References**
>
> (Azar et al., 2013) Mohammad Gheshlaghi Azar, Rémi Munos, Hilbert Kappen. Minimax PAC bounds on the sample complexity of reinforcement learning with a generative model. Machine Learning, Springer Verlag, 2013
>
> (Weng et al., 2020) Wentao Weng, Harsh Gupta, Niao He, Lei Ying, R. Srikant. The Mean-Squared Error of Double Q-Learning, NeurIPS, 2020

---

> > ### Comment · Reviewer_cHVh · 2021-08-31
> > **Response read**
> >
> > I have read the author's response  and my rating remains the same. I am not super convinced by the argument about overestimation mitigation because $Q^A$ nor $Q^B$ alone can approach  $Q^*$ too aggressively.   I agree that a conclusive non-asymptotic comparison between double Q-learning and Q-learning may be challenging. The fact that double Q-learning also achieves an optimal dependence on $\epsilon$ and $D$  is perhaps not too suprising, but what seems surprising and still remains unclear is  why double Q-learning has a worse complexity than Q-learning with a factor of O((1-\gamma)^{-2}) and requires a smaller stepsize than Q-learning from the analysis. The paper would be more insightful if the authors could shed light on the reasons for such discrepancy in the revision.

---

> > > ### Author Response · Authors · 2021-09-01
> > > **Many thanks for the further comments!**
> > >
> > > Many thanks for the further comments! We fully agree with the reviewer. We feel that formal justification of overestimation mitigation for double Q-learning may still take substantial efforts, which is our future research. Regarding the discrepancy between double Q-learning and Q-learning, our analysis indicates that this comes from the finite-time bound on $Q^A-Q^B$ (i.e., the difference between two stochastic processes, which is unique to double Q-learning) and its impact on the overall convergence error. In the revision, we can elaborate further on this. We also wish to explore in the future whether such a bound can be further improved.

---

### Official Review · Reviewer_NuYj · 2021-07-20

**Rating:** 6
**Confidence:** 2

**Summary:**

This paper analyzes the convergence rate of double Q-learning under a constant learning rate. Previous results concerning the non-asymptotic convergence of double Q-learning mainly focus on polynomial learning rate, which yields a slower convergence rate. In this work, however, the authors discuss both synchronous Q-learning and asynchronous Q-learning under a constant learning rate. This paper proves that with learning rates $\alpha \in (0, 1)$, the double Q-learning algorithm converges to an $\epsilon$-accurate global optimum with a time complexity of $\tilde{\Omega}(\frac{\ln D}{(1 - \gamma)^7 \epsilon^2})$ for the synchronous setting, and $\tilde{\Omega}(\frac{L}{(1 - \gamma)^7 \epsilon^2})$ for the asynchronous setting.

**Limitations And Societal Impact:**

The authors addressed the limitations of their work in Section 5.

**Main Review:**


This work broadens our knowledge of the non-asymptotic convergence rate of Double Q-Learning. For the synchronous setting, this paper's result improves upon Xiong et al. (2020) by $\mathcal{O}((\ln \frac{1}{1 - \gamma})^7)$. For the asynchronous setting, this paper's result improves upon Xiong et al. (2020) by at least $\mathcal{O}(L^5)$.

The main technical difficulty of using a constant learning rate lies in constructing two block-wisely decreasing bounds for the nested SAs. In this paper, the authors develop a novel analysis to directly bound both the inner and outer error dynamics and combine the two bounds to establish the finite-time error bound of the learning error.

This work extends the previous analysis of non-asymptotic convergence of double Q-learning. The article is well written and well structured.

There are, however, several points that can benefit the manuscript if tended to:

(a) The paper introduces the constant learning rate scheme and the assorted theoretical analysis for double Q-learning. However, the sense of how well the proposed learning rate and analysis are reflected in real/practical scenarios is missing.

(b) This paper provides rich theoretical analysis but very few intuitive examples. It would be great to include simple visual/numerical examples to give the audience more insights into the problem background and converging behaviors.

(c) How tight is the convergence rate in practice?

==================

Thanks for the detailed response! The authors have addressed all my concerns and I will keep my score.

**Time Spent Reviewing:**

3

---

> ### Author Response · Authors · 2021-08-10
> **Many thanks for the expert review!**
>
> **Q(a):** The paper introduces the constant learning rate scheme and the assorted theoretical analysis for double Q-learning. However, the sense of how well the proposed learning rate and analysis are reflected in real/practical scenarios is missing.
>
> **A:** Many thanks for pointing this out! We have done a quick numerical experiment, which shows that a constant learning rate indeed improves the convergence of synchronous double Q-learning over the polynomial learning rate in (Xiong et al., 2020). We will further investigate more sophisticated examples.
>
> In the revision, we will also comment on the proposed learning rate and analysis in practical performance as follows. Our theory shows that using the constant learning rate provably yields order-level better sample efficiency bound compared to the polynomial learning rate in (Xiong et al., 2020). This suggests that the practical implementation of double Q-learning should adopt a constant learning rate to achieve fast convergence.
>
> We further note that in practice often multiple advanced techniques are utilized along with double Q-learning to achieve good performance, such as variance reduction, exploration, mini-batch, etc. Our analysis can be extended to capture these variants as well. This is an interesting and important direction, which will be our future work.
>
> **Q(b):** This paper provides rich theoretical analysis but very few intuitive examples. It would be great to include simple visual/numerical examples to give the audience more insights into the problem background and converging behaviors.
>
> **A:** Thanks for the suggestions. As commented in our response to Q(a), we have conducted a numerical experiment to show the faster convergence using constant step size in comparison to the polynomial step size. We will include such an example in the revision for the suggested illustrations.
>
> **Q(c):** How tight is the convergence rate in practice?
>
> **A:** We have done a quick numerical experiment, and showed that the experimental convergence rate of double Q-learning under the constant learning rate matches our theoretical convergence rate in terms of the dependence on the learning error $\epsilon$. We will include this experiment in the revision.
>
> Analytically, by comparing with the lower bound in (Azar et al., 2013), our theoretical upper bound is tight in terms of its dependence on the major parameters: learning error $\epsilon$ and dimension $D$, whereas (Xiong et al., 2020) didn’t achieve such optimality. The only gap between our bound and the lower bound given in (Azar et al., 2013) is the dependence on $1-\gamma$, which is an interesting yet challenging topic for our future study.
>
> **Reference**
>
> (Azar et al., 2013) Mohammad Gheshlaghi Azar, Rémi Munos, Hilbert Kappen. Minimax PAC bounds on the sample complexity of reinforcement learning with a generative model. Machine Learning, Springer Verlag, 2013.

---

### Official Review · Reviewer_vEGk · 2021-08-02

**Rating:** 6
**Confidence:** 3

**Summary:**

This paper is about the analysis of both synchronous and asynchronous double Q-learning. This paper first does a quite introduction on double Q-learning, then focuses on the time complexity result for both synchronous and asynchronous double Q-learning. The time complexity result mentioned in their paper depends on the accuracy, discount factor, the number of state-action space, and covering number, and it is much better than the previous result you mention in your paper. In this paper, the authors also develop some new methods to deal with the convergence problem of double Q-learning, which makes us easier to solve the more challenging case of a constant learning rate. Since our interpretation of convergence result and time complexity of double Q-learning is still limited, I think the way you provide in the paper may give some new insight on this issue.

**Limitations And Societal Impact:**

Suggestions:
1. You mention the filtration F and the expectation H in the analysis of synchronous double Q-learning in your paper and Appendix A. I think in your paper, F and H are not formally defined. So I hope you can give a more detailed definition of F and H and explain what these notations mean in your paper.
2. I cannot catch the main idea you use when you are proving the convergence result in asynchronous double Q-learning. I think you use a quite different way in asynchronous double Q-learning compared with synchronous double Q-learning. In synchronous double Q-learning, you analyze the error per iteration. However, in asynchronous double Q-learning, you combine the error in each iteration together and treat them as a whole. What is the idea behind your paper that makes you treat synchronous and asynchronous double Q-learning differently? Could you explain more about the method you use to analyze asynchronous double Q-learning in your paper? Does that mean the dynamic of synchronous double Q-learning is quite different from that of asynchronous double Q-learning?
3. I find that the contraction property is very important in your analysis of double Q-learning. However, it is still a problem whether the function approximation method and the feedforward neural network have contraction property or under what condition it can have this property. I wonder what will happen if the contraction property is no longer satisfied in your paper. That might be helpful for the analysis of double Q-learning with function approximation.
4. I think your result may be a significant improvement in reinforcement learning. However, It is not clear how far these improvements go towards attaining the performance we see in practice, and this should be something that is explicitly addressed by the authors.


**Main Review:**

Highlights:
1. To analyze double Q-learning, the authors divide the dynamic of double Q-learning into two different stochastic approximation recursions, the outer one and the inner one. It is a natural but wise idea to treat the noise term induced by the inner stochastic approximation and other noise terms separately. The recursion formulae for both stochastic approximation recursions are very important and will be useful for the following works.
2. It is a good idea to find the similarity between the two recursions and design a template for both of them. That reveals the interrelationship between the two different stochastic approximation recursions. That also helps to make this paper more organized and easier to understand.
3. It is a wise idea that the authors use martingale and Azuma-Hoeffding inequality to deal with the noise term in the analysis of double Q-learning. It may have a wider application in dealing with the noise term. I am curious about how you come up with the idea of analyzing noise using martingale and Azuma-Hoeffding inequality.


**Time Spent Reviewing:**

8

---

> ### Author Response · Authors · 2021-08-10
> **Many thanks for the expert review!**
>
> **Q0 (Main Review, Highlights 3):** It is a wise idea that the authors use martingale and Azuma-Hoeffding inequality to deal with the noise term in the analysis of double Q-learning. It may have a wider application in dealing with the noise term. I am curious about how you come up with the idea of analyzing noise using martingale and Azuma-Hoeffding inequality.
>
> **A:** Thanks. Indeed, martingale and Azuma-Hoeffding are very useful tools here. Our novelty in this perspective lies in the construction of appropriate martingales and other techniques to facilitate such analysis. To use them in other applications, the key step will be to construct corresponding martingales.
>
> **Q1:** You mention the filtration $\mathcal{F}$ and the expectation $H$ in the analysis of synchronous double Q-learning in your paper and Appendix A. I think in your paper, $\mathcal{F}$ and $H$ are not formally defined. So I hope you can give a more detailed definition of $\mathcal{F}$ and $H$ and explain what these notations mean in your paper.
>
> **A:** Many thanks for the suggestion! The filtration $\mathcal{F}$ is defined in (7) below line 180 and $H$ is defined in Appendix A (line 488). The filtration $\mathcal{F}$ is particularly designed to facilitate the martingale construction and analysis. $H$ is defined to capture the randomness introduced by Q-estimator switching, reward, and state transitions into one variable.
>
> **Q2:** I cannot catch the main idea you use when you are proving the convergence result in asynchronous double Q-learning. I think you use a quite different way in asynchronous double Q-learning compared with synchronous double Q-learning. In synchronous double Q-learning, you analyze the error per iteration. However, in asynchronous double Q-learning, you combine the error in each iteration together and treat them as a whole. What is the idea behind your paper that makes you treat synchronous and asynchronous double Q-learning differently? Could you explain more about the method you use to analyze asynchronous double Q-learning in your paper? Does that mean the dynamic of synchronous double Q-learning is quite different from that of asynchronous double Q-learning?
>
> **A:** Many thanks for pointing this out! The main difference in the asynchronous proof is induced by the Markovian sampling. Unlike synchronous sampling where all state-action pairs are sampled at each iteration, in asynchronous sampling, only one state-action pair is sampled from a single trajectory of the underlying Markov chain. Thus, in the asynchronous case, we need to consider the dynamics of the Markov chain, and the analysis relies on the minimum stationary probability $\mu_{\min}$ and the mixing time $t_{\text{mix}}$ under the ergodicity assumption (See Eq (16) and Assumption 1). Specifically, we need to characterize the concentration of visiting state-action pairs (i.e., the probability bound on how the number of visits deviates from its mean value, see Lemma 5). Since in asynchronous sampling, only one state-action pair is sampled to update at each iteration, the decrease of the learning error (in terms of the maximum norm over all station-action pairs) can be captured only after sufficiently many iterations (in contrast to the per-iteration characterization as in the synchronous case). In this sense, the dynamics of synchronous and asynchronous double Q-learning are quite different. This is also reflected in the finite-time complexity, where additional dependence on the Markov chain through $t_{\text{mix}}$ and $\mu_{\min}$ appear in the asynchronous version (see Theorem 2).
>
> We will explain more on the proof idea of the asynchronous case in the revision.
>
> **Q3:** I find that the contraction property is very important in your analysis of double Q-learning. However, it is still a problem whether the function approximation method and the feedforward neural network have contraction property or under what condition it can have this property. I wonder what will happen if the contraction property is no longer satisfied in your paper. That might be helpful for the analysis of double Q-learning with function approximation.
>
> **A:** Good question! Indeed, function approximation remains a very challenging problem even for vanilla Q-learning. For vanilla Q-learning with linear function approximation or feedforward neural network (FNN), contraction property does not hold in general. In the literature, Q-learning under linear function approximation (Chen et al., 2021) can be shown to converge only under restrictive assumptions such as strong-convexity-like condition, boundedness of iteration, etc. Vanilla Q-learning with FNN approximation (Cai et al., 2019, Xu and Gu, 2020) can be shown to converge under similar conditions and for the overparameterized case (when FNN can be linearized). For double Q-learning with the function approximation, it is encouraging to also consider these similar assumptions.
>
>
> **Q4:** I think your result may be a significant improvement in reinforcement learning. However, It is not clear how far these improvements go towards attaining the performance we see in practice, and this should be something that is explicitly addressed by the authors.
>
> **A:** Many thanks for pointing this out! We have done a quick numerical experiment, which shows that a constant learning rate indeed improves the convergence of synchronous double Q-learning over the polynomial learning rate in (Xiong et al., 2020). This matches our theoretical result, which shows that using the constant learning rate provably yields order-level better sample efficiency bound compared to the polynomial learning rate in (Xiong et al., 2020). This suggests that the practical implementation of double Q-learning should adopt a constant learning rate to achieve fast convergence.
>
> We have done another quick numerical experiment, and showed that the experimental convergence rate of double Q-learning under the constant learning rate matches our theoretical convergence rate in terms of the dependence on the learning error $\epsilon$. In fact, analytically, our convergence rate upper bound is tight in terms of its dependence on the major parameters: the learning error $\epsilon$ and the dimension $D$, which matches the lower bound in (Azar et al., 2013). However, the bound in (Xiong et al., 2020) does not have such optimality. The only gap between our upper bound and the lower bound given in (Azar et al., 2013) lies in the dependence on $1-\gamma$, which is an interesting yet challenging topic for our future study.
>
> We also note that in practice often multiple advanced techniques are utilized along with double Q-learning to achieve good performance, such as variance reduction, exploration, mini-batch, etc. Our analysis can be extended to capture these variants as well. This is an interesting and important direction, which will be our future work.
>
> We will add these experiment results and discussions in the revision.
>
> **References**
>
> (Chen et al., 2021) Zaiwei Chen, Sheng Zhang, Thinh T. Doan, John-Paul Clarke, Siva Theja Maguluri. Finite-Sample Analysis of Nonlinear Stochastic Approximation with Applications in Reinforcement Learning, 2021
>
> (Cai et al., 2019) Qi Cai, Zhuoran Yang, Jason D. Lee, Zhaoran Wang. Neural Temporal-Difference and Q-Learning Provably Converge to Global Optima, arXiv:1905.10027, 2019.
>
> (Xu and Gu, 2020) Pan Xu and Quanquan Gu. A Finite-Time Analysis of Q-Learning with Neural Network Function Approximation, in Proc. of the 37th International Conference on Machine Learning (ICML), 2020.
>
> (Azar et al., 2013) Mohammad Gheshlaghi Azar, Rémi Munos, Hilbert Kappen. Minimax PAC bounds on the sample complexity of reinforcement learning with a generative model. Machine Learning, Springer Verlag, 2013

---

### Decision · Program_Chairs · 2021-09-28

**Decision:**

Accept (Poster)

**Comment:**

The paper offers an improved analysis of double Q-learning compared to some recent work, in particular giving better scaling in terms of the final error tolerance epsilon, the discount factor gamma, and some of the other quantities appearing in the bounds. However, there are concerns among the reviewers about the magnitude of the contribution, given that the techniques are not very novel, and that it is clear that the previous bounds compared against here are loose and can be substantially improved. Additionally, the reviewers agree that the paper does not seem to give much insight into the over-estimation problem, which is one of the main motivations for using double Q-learning, with the authors reply on this point not being very convincing. Nevertheless, on balance most reviewers lean towards acceptance, and I agree that improved bounds on the convergence time of double Q-learning are a clear contribution.


**Consistency Experiment:**

NeurIPS has a long history of experimentation. In 2014, NeurIPS ran an experiment in which 10% of submissions were reviewed by two independent committees to quantify the randomness in the review process. This year, we repeated a variant of this experiment to see how the quality of the review process has changed over time.  This paper was part of the experiment and was therefore assigned to two committees (consisting of reviewers, an Area Chair, and a Senior Area Chair) that reached independent decisions.  If both committees made the same recommendation, this recommendation was followed. If a single committee recommended acceptance, the paper was accepted (with the exception of a few cases in which the other committee identified what we considered a fatal flaw, e.g., an error in a key result).

This copy’s committee reached the following decision: **Accept (Poster)**

The other committee assigned to the paper recommended **Reject**.  You can find the other set of reviews, along with any follow up discussion with the authors here:
https://openreview.net/forum?id=sZu0b4WrElD